# Differential longitudinal establishment of human fecal bacterial communities in germ-free porcine and murine models

Nirosh D. Aluthge [1,2], Wesley A. Tom [1,3], Alison C. Bartenslager [1], Thomas E. Burkey [1], Phillip S. Miller [1], Kelly D. Heath [4], Craig Kreikemeier-Bower [4], Hatem Kittana [2,5], Robert J. Schmaltz [2], Amanda E. Ramer-Tait [2] & Samodha C. Fernando [1✉]

The majority of microbiome studies focused on understanding mechanistic relationships between the host and the microbiota have used mice and other rodents as the model of choice. However, the domestic pig is a relevant model that is currently underutilized for human microbiome investigations. In this study, we performed a direct comparison of the engraftment of fecal bacterial communities from human donors between human microbiota-associated (HMA) piglet and mouse models under identical dietary conditions. Analysis of 16S rRNA genes using amplicon sequence variants (ASVs) revealed that with the exception of early microbiota from infants, the more mature microbiotas tested established better in the HMA piglets compared to HMA mice. Of interest was the greater transplantation success of members belonging to phylum Firmicutes in the HMA piglets compared to the HMA mice. Together, these results provide evidence for the HMA piglet model potentially being more broadly applicable for donors with more mature microbiotas while the HMA mouse model might be more relevant for developing microbiotas such as those of infants. This study also emphasizes the necessity to exercise caution in extrapolating findings from HMA animals to humans, since up to 28% of taxa from some donors failed to colonize either model.

[1] Department of Animal Science, University of Nebraska-Lincoln, Animal Science Complex, 3940 Fair St., Lincoln, NE 68583-0908, USA. [2] Department of Food Science and Technology, Food Innovation Center, University of Nebraska–Lincoln, 1901 N 21st St., Lincoln, NE 68588-6205, USA. [3] School of Biological Sciences, University of Nebraska–Lincoln, Manter Hall, 1104 T St., Lincoln, NE 68588-0118, USA. [4] Institutional Animal Care Program, University of Nebraska–Lincoln, 110 Mussehl Hall, 1915 N 38th St., Lincoln, NE 68653-0720, USA. [5]Present address: Veterinary Medical Diagnostic Laboratory (VMDL) at University of Missouri (MU), 901 E Campus Loop, Columbia, MO 65211, USA. ✉email: samodha@unl.edu

The human gut microbiota has been intensively studied during the past couple of decades, and much light has been shed on the importance of this microbial community in influencing the health and well-being of humans and animals[1]. A broad range of human disease conditions ranging from immune and metabolic pathologies to behavioral and psychological conditions[2–4] have been linked to aberrations in gut microbiota composition. However, most of the apparent links made so far between the gut microbiota and human health conditions have been limited to correlations. Consequently, the role of the microbiota as a driver of pathologies has been difficult to establish[5].

To move beyond correlations and identify the causality and mechanistic role of the microbiota in influencing human health, researchers have turned to human microbiota-associated (HMA) animal models as surrogates for human subjects. To date, rodents are the most widely-used HMA animal model, but HMA pigs (Sus scrofa domesticus) also represent a highly-relevant model system for studying host-microbiota interactions[6–16]. The 'humanization' process involves the inoculation of germ-free (GF) animals with human fecal matter to establish human gut microbial communities in these animal models while controlling for many confounding factors such as environment, diet, and host genetics. A number of human-to-animal fecal transplantation studies using recipient GF mice appear to have been successful in recapitulating human donor phenotypes such as development of obesity[17], amelioration of asthma symptoms[18], and manifestation of autistic behaviors[19], although a recent systematic review questioned the ability to transfer human pathologies to HMA rodent models[20]. However, interspecies gut microbiota transplantations are known to cause shifts in the original donor microbiota composition as it establishes and adapts to the new gut environment of the recipient species[21]. Unfortunately, the extent to which the composition of a human microbiota represents that of the donor following transplant into a recipient animal is not always assessed in studies using HMA animal models[22,23]. If microbial establishment is evaluated in the recipient mouse or pig, then such comparisons are often limited to only describing higher

taxonomic levels such as phylum, class, or genus[8,24] or measures of global microbiota structure such as alpha and beta diversity[22]. In addition, many reports using HMA animals do not evaluate whether the transplanted microbiota is stably maintained over time within the new host[22]. This lack of reporting on the fine-scale differences of microbiotas before and after transplant highlights an important caveat of working with HMA animal models—the ability of animal models to establish and maintain a human gut microbiota that closely resembles the donor over time is currently not well-characterized.

The objective of this study was to evaluate the establishment of human fecal bacterial communities derived from multiple human donors with varying levels of microbiota diversity in porcine and murine models. By transplanting fecal microbiotas from the same human donors into GF piglets and mice and maintaining both species under identical dietary and similar environmental conditions, we were also able to directly compare the ability of HMA porcine and murine models to harbor and maintain 'human-like' gut bacterial communities over time. The results from the current study suggest that human fecal bacterial communities from donors with more mature microbiotas establish more successfully in the HMA porcine model while the simpler infant microbial communities favor the HMA murine model. Specifically, members of the phylum Firmicutes had greater success in colonizing the HMA piglets compared to the HMA mice. These findings point to the usefulness of both of these animal models for human microbiome studies with the HMA porcine model potentially having a broader scope.

## Results

**Identifying core microbiotas in the human donors.** To compare the establishment of human fecal bacterial communities in HMA mice and piglets, we inoculated GF mice and piglets maintained in gnotobiotic isolators with fecal matter from four separate human donors. The donors selected had diverse microbial communities (Fig. 1) and represented different stages of human development (see "Methods" for donor information). All animals in a given isolator (for both mice and piglets) were inoculated with the inocula obtained from a single donor. Both recipient species of animals were inoculated twice during the study—the initial round of inoculations were performed after weaning and the second round of inoculations occurred two weeks after the first round of inoculations. All inocula were prepared at the same time under the same conditions and both mice and piglets were fed the exact same sterile solid diet.

To compare the engraftment of donor microbiota in the two HMA animal models, the establishment of amplicon sequence variants (ASVs) identified in the donor inocula was evaluated. To reduce variation in composition due to sequencing depth, a 'core' bacterial community was defined for each donor by identifying ASVs that were found in all aliquots of the pooled donor inoculum used to inoculate the corresponding HMA animals (see "Methods" for details on donor inoculum preparation). There were a total of 4 inoculum aliquots per donor – two used for inoculating the GF mice and two used for inoculating the GF piglets. All 4 inoculum aliquots were sequenced separately. For Donor_1, one of the inoculum aliquots used to inoculate the corresponding GF piglets did not have sufficient material for DNA extraction, which resulted in only three inoculum aliquots from this donor for subsequent analysis. Twenty-six, 76, 140, and 134 core ASVs that fit the above criteria were identified from Donor_1, Donor_2, Donor_3, and Donor_4, respectively. As a percentage of the total number of reads, the core ASVs identified accounted for

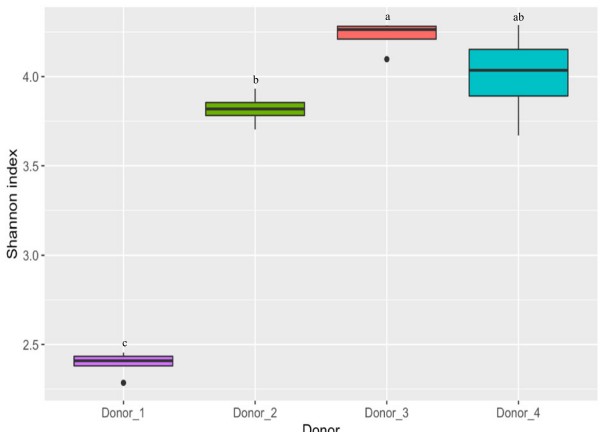

**Fig. 1 Box-whisker plots comparing the alpha diversity of the inoculum aliquots among the different donors using the Shannon index.** Statistical comparisons were performed using the Wilcoxon rank-sum test. Boxes with different letters indicate statistically significant differences ($p < 0.05$). The box represents the 25th and 75th percentiles as the interquartile range (IQR) and the short black line represents the median. The whiskers represent the minimum and maximum values. Outliers are shown as black dots. $n = 3$ inoculum aliquots for Donor_1 and $n = 4$ inoculum aliquots or Donor_2, Donor_3, and Donor_4.

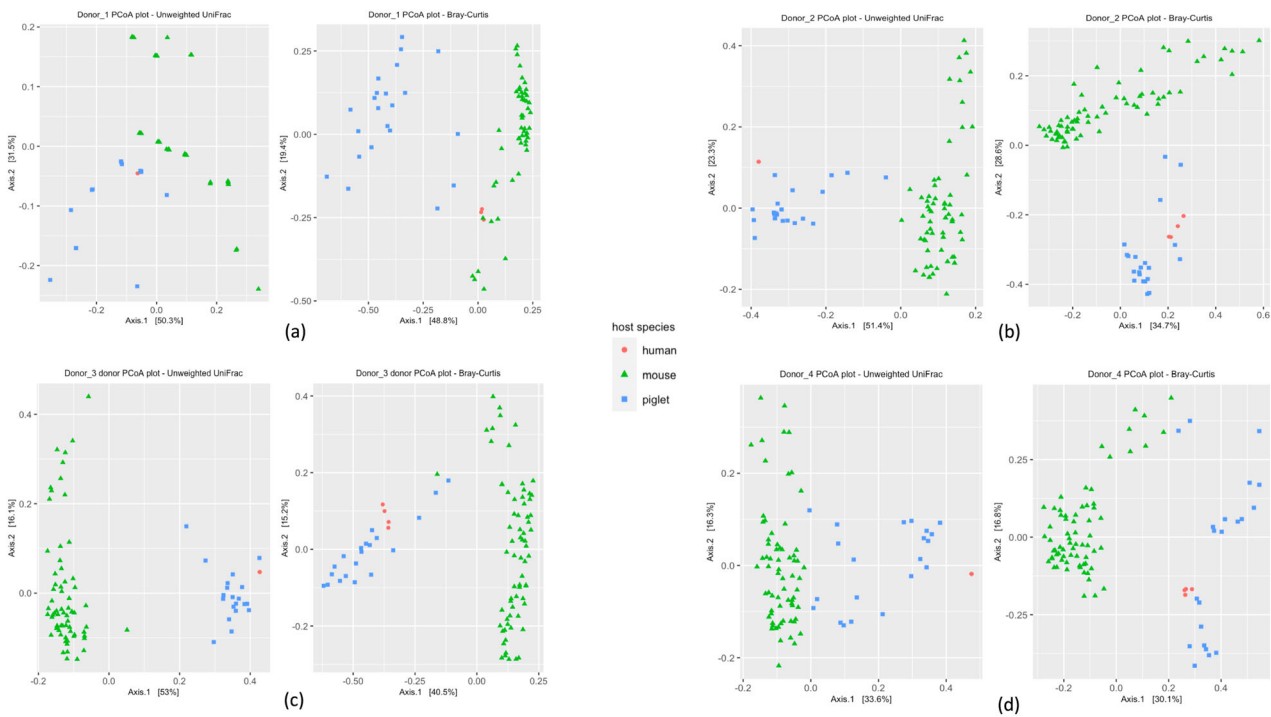

**Fig. 2 Principal Coordinate Analysis (PCoA) plots using unweighted UniFrac and Bray-Curtis distances for the bacterial communities in each of the four donor inocula and the fecal samples from the corresponding HMA animal models.** (**a**) Donor_1, (**b**) Donor_2, (**c**) Donor_3, (**d**) Donor_4. $n = 13$ (3 piglets/donor, with the exception $n = 4$ for Donor_1, 2 days post inoculation) and $n = 37$ mice (Donor_1 = 7, Donor_2 = 10, Donor_3 = 10, Donor_4 = 10). Red circles, human donor inocula; green triangles, HMA mouse fecal samples; blue squares, HMA piglet fecal samples.

99%, 96%, 98%, and 96% of the total reads in the 4 donor inocula used in the study. The taxonomy and relative abundance information of the core ASVs identified for each donor across the animal models is provided in Supplementary Data 1. Since core ASVs identified are found in all inoculum aliquots of a given donor and represent the major portion of the reads generated (96–99%), we reasoned that these taxa can be considered as the major taxa within the human donors having the highest probability of colonizing each animal model. With the exception of alpha diversity analysis, the remainder of the results presented are focused on the core ASVs identified.

**Global diversity patterns in the HMA animal models**. To evaluate global bacterial community differences, we compared the establishment of the core microbiotas of the corresponding human donors in the two HMA animal models using Principal coordinate analysis (PCoA) plots based on unweighted UniFrac and Bray-Curtis distances (Fig. 2). For Donor_1, the Bray-Curtis distance-based analysis revealed the donor inoculum aliquots to cluster more closely to the HMA mouse fecal samples than the HMA piglet fecal samples (Fig. 2a). However, for Donor_2, Donor_3, and Donor_4, the donor inoculum aliquots were more similar to the HMA piglet fecal samples than the HMA mouse fecal samples (Fig. 2b–d). As previously reported[25], we also compared microbial community engraftment using UniFrac distances and Bray–Curtis distances (Fig. 3), which further corroborated the results from the beta diversity analysis.

To evaluate the weekly variation in alpha diversity of recipient microbiotas, we compared the alpha diversity of the HMA animals with that of their respective human donors at each time point (Fig. 4). Since alpha diversity analyses are sensitive to the presence/absence of low prevalence taxa, we considered all ASVs (not only core ASVs) of a given donor for this analysis. Donor_1-HMA mice maintained a stable alpha diversity over time, with

only the fecal samples from week 4 showing a Shannon index significantly different ($p = 0.018$) from that of Donor_1 which represents an infant microbiota. However, greater variation in alpha diversity was observed in Donor_1-HMA piglet fecal samples with days 2, 7, 14, 21, and 40 post-inoculation yielding significantly different Shannon indexes compared to the Donor_1 inoculum aliquots (Fig. 4a). Diversity estimates for Donor_2-HMA and Donor_3-HMA piglet fecal samples were maintained similar to that of the estimates for the human donors throughout the study (Fig. 4b, c). However, significantly lower Shannon index values for Donor_2-HMA and Donor_3-HMA mouse fecal samples were observed compared to the donor inoculum aliquots. Donor_4-HMA piglet fecal microbiota displayed notable variations in alpha diversity throughout the study period (Fig. 4d). The Donor_4-HMA mouse fecal samples did not show much variation in alpha diversity between sampling time points although the Shannon indexes were significantly lower relative to the donor inoculum aliquots.

**Establishment of core donor ASVs in the two HMA animal models**. Heatmaps (Supplementary Figs. 1–4) and chord diagrams (Fig. 5) were generated to visualize the establishment of the core donor ASVs in HMA mice and piglets, which revealed notable differences in their success of engraftment in the two species. Notably, certain ASVs only established in one of the two animal models, while some others did not establish in either. With the exception of Donor_1, more core ASVs found in the human donors were identified in the HMA piglets compared to the HMA mice.

To determine how successfully the core ASVs found in human donors established and persisted in the corresponding HMA animal models, we defined 'colonization criteria', which were established based on the number of animals and sampling time points in which a donor ASV was detected. If a core donor ASV

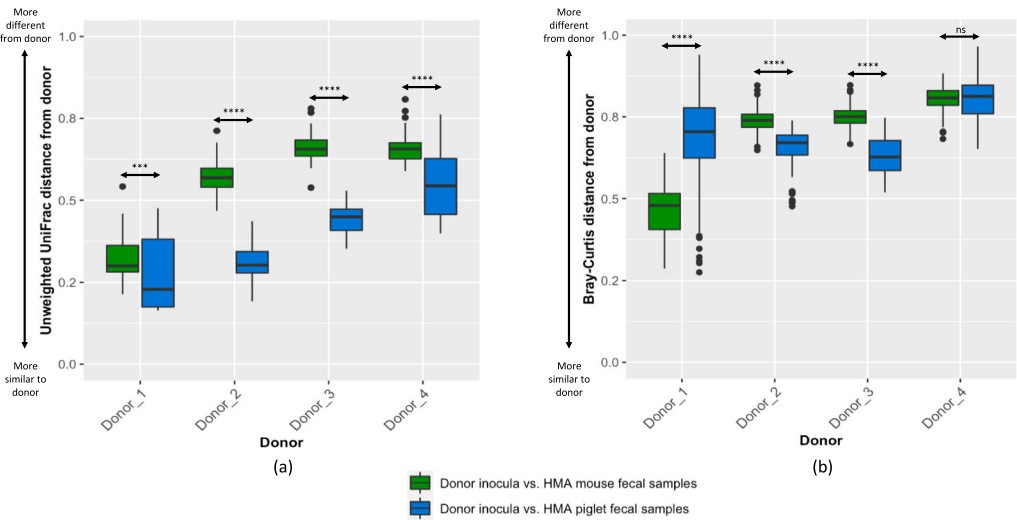

**Fig. 3 Comparison of similarity of HMA animal fecal bacterial communities and corresponding human donor bacterial communities.** (**a**) Unweighted UniFrac distances and (**b**) Bray–Curtis distances. The box represents the 25th and 75th percentiles as the interquartile range (IQR) and the short black line represents the median. The whiskers represent the minimum and maximum values. Outliers are shown as black dots. Statistical comparisons were performed using the Wilcoxon rank-sum test. $***p < 1 \times 10^{-5}$; $****p < 1 \times 10^{-10}$; ns–not significant. $n = 13$ (3 piglets/donor, with the exception $n = 4$ for Donor_1, 2 days post inoculation) and $n = 37$ mice (Donor_1 = 7, Donor_2 = 10, Donor_3 = 10, Donor_4 = 10).

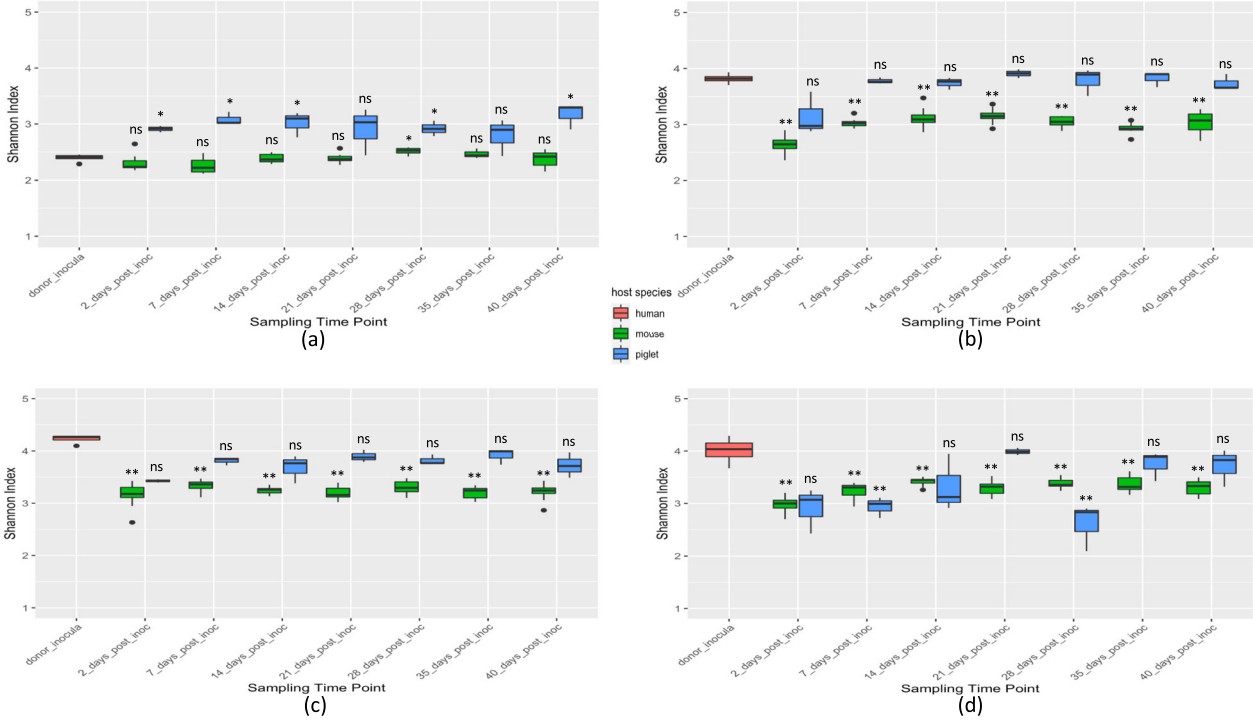

**Fig. 4 Box-whisker plots depicting the variation in alpha diversity between the donor inocula and the HMA animal fecal samples at each sampling time point.** (**a**) Donor_1, (**b**) Donor_2, (**c**) Donor_3, and (**d**) Donor_4. The box represents the 25th and 75th percentiles as the interquartile range (IQR) and the short black line represents the median. The whiskers represent the minimum and maximum values. Outliers are shown as black dots. Statistical comparisons are based on the Wilcoxon rank-sum test. $*p < 0.05$; $**p < 0.01$; ns–not significant. $n = 13$ (3 piglets/donor, with the exception $n = 4$ for Donor_1, 2 days post inoculation) and $n = 37$ mice (Donor_1 = 7, Donor_2 = 10, Donor_3 = 10, Donor_4 = 10).

was detected in at least one fecal sample of a given HMA animal model, then this ASV was considered as a 'colonizer.' However, if a donor ASV was detected in fecal samples of >50% of the animals in at least 4 of the 7 sampling time points, then this ASV was considered as a 'persistent colonizer' for that animal model (Table 1 and Supplementary Data 2). Relative abundance and taxonomic information related to persistent colonizers identified

following these criteria for each donor and animal model are provided in Supplementary Data 3.

For Donor_1, a single ASV out of the five core ASVs that were not detected in the HMA mice was detected in the HMA piglets. Similarly, 2 ASVs of the 6 core Donor_1 ASVs that were not detected in the HMA-piglets were detected in the HMA mice. Four ASVs (15%) failed to colonize either animal model.

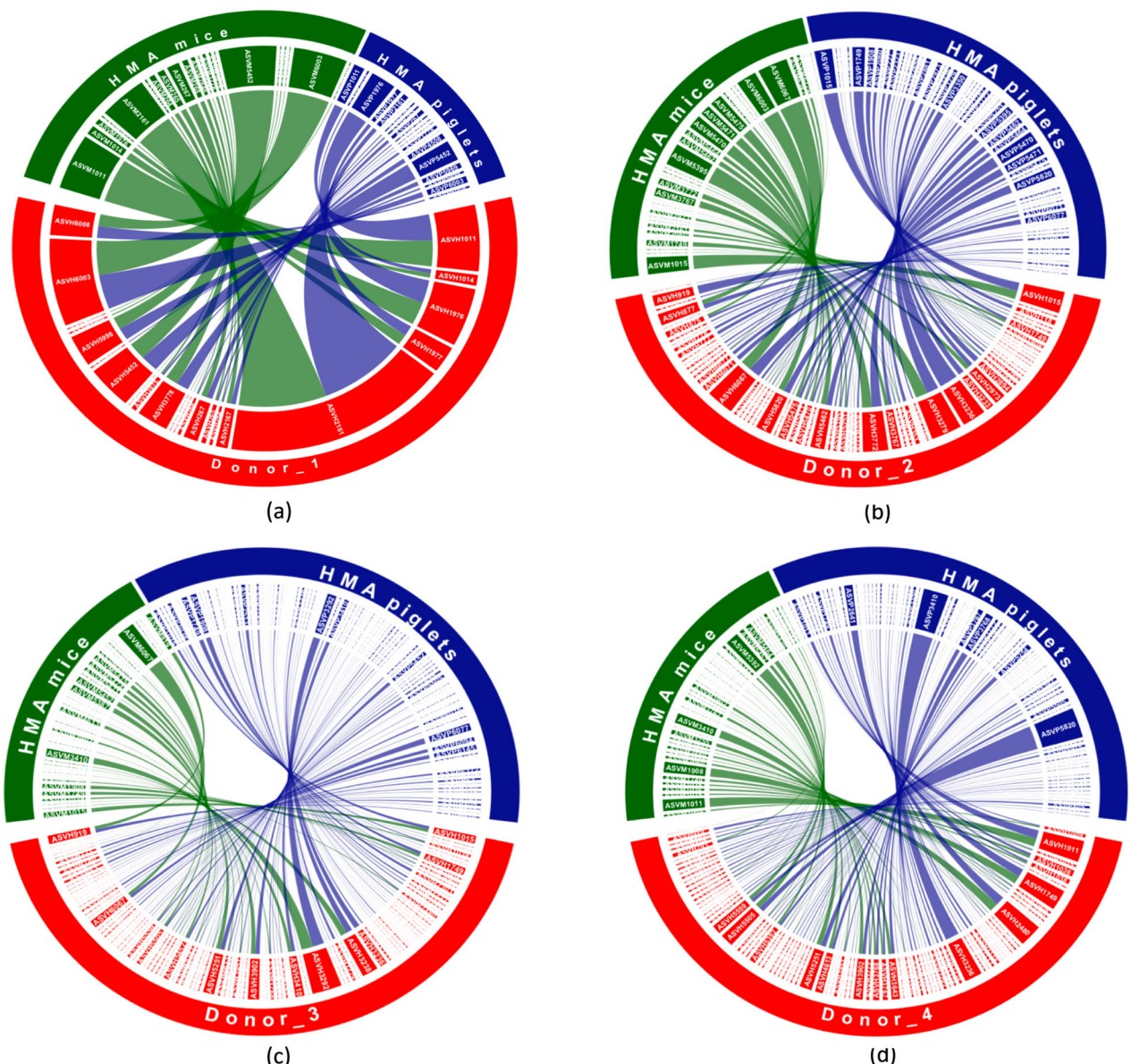

**Fig. 5 Chord diagram representing the colonization patterns of core donor ASVs in the two HMA animal models.** (**a**) Donor_1, (**b**) Donor_2, (**c**) Donor_3, (**d**) Donor_4. Each sector represents an ASV and the size of the sector corresponds to the mean relative abundance of that ASV in the host species. Links indicate which core donor ASVs established in each animal model and the widths at each end of a link are proportional to the abundance of that ASV in the respective host species. Core donor ASVs which failed to colonize both HMA animal models are not shown. ASVs have been designated according to the following example: ASV_1 is ASVH1 in human donors, ASVM1 in HMA mice, and ASVP1 in HMA piglets. $n = 13$ (3 piglets/donor, with the exception $n = 4$ for Donor_1, 2 days post inoculation) and $n = 37$ mice (Donor_1 = 7, Donor_2 = 10, Donor_3 = 10, Donor_4 = 10).

**Table 1 Summary of the establishment of core donor ASVs in the two HMA animal models for each donor.**

| Donor | Number of core donor ASVs | HMA animal model | Number of colonizers | Number of persistent colonizers | Average read depth for fecal samples |
|---|---|---|---|---|---|
| Donor_1 | 26 | Mice | 21 (80.8%) | 16 (61.5%) | 45,238 |
| | | Piglets | 20 (76.9%) | 15 (57.7%) | 37,341 |
| Donor_2 | 76 | Mice | 37 (48.7%) | 21 (27.6%) | 47,814 |
| | | Piglets | 65 (85.5%) | 43 (56.6%) | 44,321 |
| Donor_3 | 140 | Mice | 66 (47.1%) | 31 (22.1%) | 45,216 |
| | | Piglets | 106 (75.7%) | 69 (49.3%) | 39,972 |
| Donor_4 | 134 | Mice | 55 (41.0%) | 28 (20.9%) | 45,833 |
| | | Piglets | 92 (68.6%) | 38 (28.3%) | 34,309 |

For Donor_2 and Donor_3, a higher proportion of core donor ASVs established in the piglets compared to the mice (Table 1). Twenty-eight of the 39 Donor_2 core ASVs that failed to colonize the mice colonized the piglets. The 11 ASVs that were not detected in the piglets were also absent in the mice. Only 3 out of the 34 Donor_3 core ASVs that failed to colonize the HMA piglets colonized the HMA mice. However, 43 out of the 74 core donor ASVs that failed to colonize the HMA mice colonized the HMA piglets. Eleven (14%) and 31 (22%) core ASVs failed to colonize either animal model for Donor_2 and Donor_3, respectively.

For Donor_4, Forty-one out of the 79 core ASVs that did not colonize the HMA mice colonized the HMA piglets. Only 4 out of the 42 ASVs that failed to colonize the HMA piglets colonized the HMA mice. Thirty-eight (28%) core Donor_4 ASVs failed to establish in either animal model. The taxonomic classifications of the core ASVs in each donor that failed to colonize both HMA animal models are provided in Supplementary Data 4.

**Establishment of common ASVs found among donors in the two HMA animal models.** With the assumption that ASVs common to all donors may represent part of the autochthonous microbes in the human gut, we evaluated the establishment of such ASVs in the two HMA models. However, Donor_1 was not considered for this analysis as the Donor_1 microbiome is from an infant, and infants are known to have unstable and highly dynamic gut microbiota compositions[26–28]. We identified 27 core donor ASVs (Supplementary Data 5) that were shared by all 12 inoculum aliquots originating from Donor_2, Donor_3, and Donor_4 (4 aliquots for each donor). Twenty-four of these shared ASVs were classified as members of the phylum Firmicutes and 2 were classified as members of phylum Bacteroidetes. The remaining ASV belonged to the phylum Desulfobacterota (Supplementary Data 5). The 27 common ASVs identified were compared between the two HMA models in terms of how many 'colonizers' and 'persistent colonizers' were present. The results of this analysis (Supplementary Data 6) revealed that, for all three donors, the 27 shared core ASVs established more efficiently in the HMA piglets compared to the HMA mice. Specifically, 24 of these 27 ASVs (89%) had established in the Donor_2-HMA piglets, while 25 (93%) and 19 (70%) had colonized the HMA piglets inoculated with Donor_3 and Donor_4, respectively. In contrast, only 9 (33%), 15 (55%), and 10 (37%) of these ASVs had colonized the HMA mice for Donor_2, Donor_3, and Donor_4, respectively. In terms of persistent colonizers, 16 (59%), 20 (74%), and 6 (22%) of the 27 common ASVs were identified as persistent colonizers of the piglets inoculated with Donor_2, Donor_3, and Donor_4, respectively. For the same donors, 5 (18%), 6 (22%), and 5 (18%) ASVs were persistent colonizers of the mice (Supplementary Data 6).

**Members of the phylum Firmicutes establish in the HMA piglets with greater success compared to the HMA mice while Bacteroidetes successfully colonize both animal models.** We compared core donor taxa establishment at phylum, family, and genus levels between the two HMA animal models for each donor (Supplementary Data 7). At the phylum level, the most notable differences between the two animal models were observed for the phylum Firmicutes. Donor_2 had 58 core ASVs belonging to Firmicutes, and only 21 (36%) of these ASVs established in the HMA mice; however, 48 (83%) colonized the HMA_piglets. Out of the 48 core Firmicutes ASVs that established in the HMA piglets, 31 (65%) were persistent colonizers, while 10 of the 21 (48%) Firmicutes ASVs persistently colonized the HMA mice (Supplementary Data 2). The Donor_3 inocula contained 97 core

Firmicutes ASVs, 71 (73%) of which established in HMA piglets and 31 (32%) established in the HMA mice. Only 6 out of the 31 (19%) core Firmicutes ASVs colonizing the HMA mice were persistent colonizers, whereas 44 of the 71 (62%) Firmicutes ASVs persistently colonized the HMA piglets. Similarly, out of the 95 core Firmicutes ASVs found in the Donor_4 inocula, 21 (22%) colonized the HMA mice and 57 (60%) colonized the HMA piglets. Eight of the 21 (38%) Firmicutes core ASVs that established in the HMA mice were persistent colonizers, while 15 of the 57 (26%) Firmicutes ASVs persistently colonized the HMA piglets. Phylum Firmicutes was represented in 18 ASVs and was the predominant phylum in the Donor_1 core with a mean abundance of 62.77% (SD = 5.18%). Fourteen and thirteen of these Firmicutes ASVs colonized the HMA mice and HMA piglets, respectively, with mean relative abundances of 40.5% (SD = 11.15%) and 10.72% (SD = 7.6%), respectively (Supplementary Data 7).

The differential establishment of Firmicutes in the two HMA animal models was observed at the family level by members of Lachnospiraceae and Ruminococcaceae (Supplementary Data 7). Of the 34 Donor_2 core Lachnospiraceae ASVs, only 11 (32%) were detected in the HMA mice while 28 (82%) were detected in the HMA piglets. Sixteen of the 28 (57%) Lachnospiraceae ASVs detected in the piglets were persistent colonizers, while 6 of the 11 (54%) Lachnospiraceae ASVs that established in the mice were persistent colonizers (Supplementary Data 2). Similarly, the Donor_3 core consisted of 40 Lachnospiraceae ASVs, 15 (37%) of which colonized the HMA mice and 31 (77%) which colonized the HMA piglets. Only 3 out of the 15 (20%) Lachnospiraceae ASVs colonizing the HMA mice were persistent colonizers. In contrast, 20 out of the 31 Lachnospiraceae ASVs (64%) were persistent colonizers in HMA piglets. The Donor_4 core contained 40 Lachnospiraceae ASVs; only 7 of these ASVs (17%) established in the HMA mice, whereas 22 (55%) established in the HMA piglets. Two core Lachnospiraceae ASVs (29%) persistently colonized the HMA mice, while 4 out of the 22 (18%) core Lachnospiraceae ASVs colonizing the HMA piglets were persistent colonizers (Supplementary Data 2). Similar to Lachnospiraceae, core donor ASVs belonging to the family Ruminococcaceae were poorly established in the HMA mice relative to the HMA piglets (Supplementary Data 7). Donor_2 contained 6 core ASVs belonging to Ruminococcaceae. All 6 of these ASVs (100%) successfully colonized the HMA piglets, while only 1 (17%) colonized the HMA mice. Four of the 6 (67%) core Ruminococcaceae ASVs that successfully colonized the HMA piglets were identified as persistent colonizers (Supplementary Data 2). Out of the 17 core ASVs belonging to Ruminococcaceae found in Donor_3, 12 (71%) colonized the HMA piglets, while only 6 (35%) colonized the HMA mice. One of these 6 core Ruminococcaceae ASVs (17%) persistently colonized the mice. However, 9 out of the 12 (75%) Donor_3 core Ruminococcaceae ASVs that established in the piglets were persistent colonizers. Donor_4 consisted of 14 core Ruminococcaceae ASVs but only 4 (29%) colonized the HMA mice. In contrast, 12 (86%) core Ruminococcaceae ASVs colonized the piglets. Three of the 4 (75%) core Ruminococcaceae ASVs persistently colonized the mice, whereas 5 out of the 12 (42%) core Ruminococcaceae ASVs persistently colonized the piglets (Supplementary Data 2).

The Firmicutes family Christensenellaceae, which has recently been identified as the most heritable bacterial family in the human microbiota and also related to health benefits[29,30], successfully colonized the HMA piglets but was not detected in the HMA mice (Supplementary Data 7). Specifically, Christensenellaceae ASVs were identified as core members only in Donor_3 and Donor_4 inocula, with 6 core ASVs in Donor_3 and 7 core ASVs in Donor_4. Out of the 6 core Christensenellaceae ASVs in

Donor_3, 4 established in the HMA piglets–3 of these were persistent colonizers. In addition, 4 out of the 7 core Christensenellaceae ASVs in Donor_4 established in the HMA piglets. A single core Christensenellaceae ASV persisted in these HMA piglets (Supplementary Data 2).

In contrast to the notable differences in colonization efficiency of members of the phylum Firmicutes between the HMA mice and the piglets, ASVs belonging to Bacteroidetes successfully colonized both animal models (Supplementary Data 7). All 3 of the core Bacteroidetes ASVs identified in Donor_1 persistently colonized both animal models (Supplementary Data 2), although their abundances were much higher in the HMA mice (mean = 35.56%, SD = 14.42%) compared to the HMA piglets (mean = 9.74%, SD = 6.98%). Similarly, all 8 core Bacteroidetes ASVs of Donor_2 persistently colonized both HMA animals (Supplementary Data 2) and at higher relative abundances compared to the donor (Supplementary Data 7). Twenty-one of the 25 core Bacteroidetes ASVs of Donor_3 established in both animal models, with 18 (85.7%) and 16 (76.2%) of these ASVs persistently colonizing the HMA mice and HMA piglets, respectively. Higher relative abundances for these Bacteroidetes ASVs were observed in the HMA mice compared to both the donor and the HMA piglets (Supplementary Data 7). Finally, of the 25 core Bacteroidetes ASVs detected in Donor_4, 21 (84%) and 23 (92%) colonized the HMA mice and piglets, respectively, although their relative abundances were much lower compared to the donor (Supplementary Data 7). Fourteen of the 21 (67%) Bacteroidetes ASVs colonizing the mice were persistent colonizers, while 13 of the 23 (56%) Bacteroidetes ASVs persistently colonized the piglets (Supplementary Data 2).

Together, our results demonstrate that a greater number of ASVs belonging to the phylum Firmicutes, especially those of the families Lachnospiraceae, Ruminococcaceae, and Christensenellaceae, from the human donors established and persisted in the HMA piglets compared to the HMA mice. Our results also indicate that members of the phylum Bacteroidetes successfully colonized both animal models, although their relative abundances were higher in the HMA mice compared to the HMA piglets.

**Bacteroides and Bifidobacterium members successfully colonize both animal models**. The genera Bacteroides and Bifidobacterium are key members of the human gut microbiota and were identified as part of the core microbiotas of all four donors in this study (Supplementary Data 1). The Donor_1 core included a single Bifidobacterium ASV classified as Bifidobacterium bifidum (mean abundance = 4.5%, SD = 3.47%), which colonized both animal models at much lower relative abundances compared to the donor with means of 0.81% (SD = 0.71%) and 0.84% (SD = 1.17%) for the HMA mice and HMA piglets, respectively. This Bifidobacterium bifidum ASV also persistently colonized both HMA animal models (Supplementary Data 3). The Donor_2 core contained 3 Bifidobacterium ASVs, with all 3 ASVs colonizing the HMA piglets and 2 out of the 3 ASVs colonizing the HMA mice. The Donor_3 and Donor_4 cores contained 1 and 3 Bifidobacterium ASVs, respectively, all of which established in both HMA animal models. The single core Bifidobacterium ASV identified for Donor_3 did not persist in either animal model. Only 1 of the 3 core Bifidobacterium ASVs of Donor_4 persistently colonized the mice while all 3 ASVs persistently colonized the piglets (Supplementary Data 2).

All core Bacteroides ASVs found in the donors successfully colonized both animal models (Supplementary Data 3). Donor_1 core contained 3 Bacteroides ASVs at a mean abundance of 18.05% (SD = 0.34%). Two of these Bacteroides ASVs were assigned to the species B. vulgatus while the remaining ASV was

classified as B. uniformis (Supplementary Data 1). These Bacteroides ASVs bloomed in the HMA mice with a mean abundance of 35.56% (SD = 14.42%). In contrast, these same ASVs reached a mean relative abundance of only 9.74% (SD = 6.98%) in the HMA piglets. All 3 of these Bacteroides ASVs persistently colonized both animal models (Supplementary Data 2). Donor_2 contained 6 core Bacteroides ASVs (mean abundance = 11.37%, SD = 1.42%), which established at higher abundances in both HMA animal models—26.86% (SD = 7.23%) and 22.80% (SD = 6.09%) for the HMA mice and piglets, respectively (Supplementary Data 7). In addition, all 6 of these ASVs persisted in both groups of animals (Supplementary Data 2). The 7 core Bacteroides ASVs in Donor_3 (mean abundance = 6.51%, SD = 0.67%) established in the HMA piglets at similar levels but with greater variation (mean abundance = 7.01%, SD = 8.67%). However, in the HMA mice, the abundances were much greater (mean abundance = 18.9%, SD = 6.34%) compared to both the donor and the HMA piglets. All 7 ASVs persistently colonized the HMA mice, while 6 out of the 7 persisted in the HMA piglets. The Donor_4 core contained 7 Bacteroides ASVs (mean abundance = 7.13%, SD = 0.62%) and these ASVs colonized the HMA mice at comparable relative abundances (mean = 9.52%, SD = 4.53%). The relative abundances for the same Bacteroides ASVs were much lower in the HMA piglets (mean = 3.01%, SD = 4.16%). Six and 4 of these 7 core Bacteroides ASVs persistently colonized the HMA mice and piglets, respectively (Supplementary Data 2).

**Establishment of ASVs classified as Faecalibacterium prausnitzii, Roseburia and Ruminococcus bromii in the HMA animal models**. Previous studies have demonstrated Faecalibacterium prausnitzii and Roseburia species to be functionally important members of the human gut microbiota and potential biomarkers of gut health[31,32]. We therefore evaluated the establishment and persistence of these ASVs in the two HMA animal models. Donor_2, Donor_3, and Donor_4 had core ASVs which were classified as Faecalibacterium prausnitzii (Supplementary Data 1). The Faecalibacterium prausnitzii (ASV_3233) of the Donor_2 core was detected only in the HMA piglets but at a lower relative abundance. Two ASVs classified as Faecalibacterium prausnitzii (ASV_3233 and ASV_3238) were part of the Donor_3 core. While ASV_3238 had colonized both HMA animal models, ASV_3233 only colonized the HMA piglets (Supplementary Data 1). Donor_4 contained three ASVs classified as Faecalibacterium prausnitzii (ASV_3233, ASV_3238, and ASV_3239). All three ASVs were detected in the HMA piglets, while none were detected in the HMA mice (Supplementary Data 1). Thus, all the core Faecalibacterium prausnitzii ASVs from the different donors colonized the HMA piglets, albeit at lower relative abundances. However, only ASV_3233 found in the Donor_3-HMA piglets was a persistent colonizer (Supplementary Data 3).

The butyrate-producing human gut commensal genus Roseburia was represented among the core ASVs of Donor_2, Donor_3, and Donor_4 inocula (Supplementary Data 1). All core Roseburia ASVs found in each donor (2, 3, and 2 ASVs for Donor_2, Donor_3, and Donor_4, respectively) colonized the corresponding HMA piglets. The relative abundances of these ASVs were comparable or higher to that of the donor, with the exception of ASV_6057 found in the Donor_2-HMA and Donor_4-HMA piglets (Supplementary Data 1). Furthermore, apart from the 2 core Roseburia ASVs of Donor_4, the remaining core Roseburia ASVs that established in the HMA piglets were persistent colonizers (Supplementary Data 2). In contrast, the only core Roseburia ASVs that established in the HMA mice were 2 core Roseburia ASVs found in Donor_3. These ASVs colonized

at very low abundances (mean abundance = <0.01%, SD = 0.01%) in the HMA mice. Across donors, the main species of *Roseburia* identified in the core microbiotas were *R. inulinivorans* and *R. intestinalis* (Supplementary Data 1).

The prominent resistant starch degrading gut symbiont *Ruminococcus bromii* has also been proposed as a keystone species within the human gut microbiota[33]. *R. bromii* (ASV_3350) was identified in the core microbiotas of Donor_2 and Donor_4 (Supplementary Data 1). *R. bromii* had only engrafted successfully in the corresponding HMA piglets and at higher relative abundances than in the human donors (Supplementary Data 1). Moreover, this ASV had persistently colonized both Donor_2-HMA and Donor_4-HMA piglets (Supplementary Data 3).

**The ability of persistent colonizers to maintain donor-like abundances in the HMA animal models over time**. To determine if the persistent colonizers identified in the HMA animals were capable of maintaining abundances similar to the human donors over time, we compared the relative abundances of the persistent colonizer ASVs established in the two HMA animal models to their abundances in the respective human donors at each sampling time point. For persistent colonizer ASVs colonizing each HMA animal model, we determined the number of time points in which the relative abundance of each of the ASVs were significantly different from the corresponding human donor inocula. We considered a persistent colonizer ASV as having maintained a 'donor-like' relative abundance if that ASV's relative abundance in a given animal model was not significantly different from the corresponding human donor in at least 4 out of the 7 sampling time points. Supplementary Data 8 and 9 provide the relative abundance and taxonomic information for all persistent colonizer ASVs with donor-like relative abundances with respect to each HMA animal model.

Based on the above criterion, we observed 7 out of 16 (44%) and 8 out of 15 (53%) of the persistent colonizer ASVs to maintain donor-like relative abundances for Donor_1 in the HMA mice and HMA piglets, respectively. *Bifidobacterium bifidum* (ASV_3778) was able to maintain a donor-like relative abundance in the HMA mice but not in the HMA piglets (Supplementary Data 9). For Donor_2, the HMA mice contained only 2 out of the 21 (9%) persistent colonizer ASVs with donor-like abundances. The HMA piglets contained 15 out of 43 (35%) persistent colonizer ASVs with donor-like relative abundances. For Donor_2, members of the genera *Bacteroides*, *Blautia*, and *Roseburia* were among the ASVs with donor-like abundances in the HMA piglets (Supplementary Data 9). For Donor_3, 40 out of the 69 (58%) persistent colonizers maintained donor-like abundances in the HMA piglets while only 6 out of 31 (19%) persistent colonizer ASVs identified in the HMA mice were able to do so. Five out of the 6 persistent colonizers with donor-like abundances identified for Donor_3 in HMA mice were Bacteroidetes members. Bacteroidetes and Firmicutes were widely represented among the persistent colonizers with donor-like abundances for Donor_3 in HMA piglets (Supplementary Data 9). For Donor_4, 7 out of 28 (25%) and 16 out of 38 (42%) persistent colonizers were maintained at donor-like relative abundances in the HMA mice and HMA piglets, respectively.

Altogether, these findings suggest that only a small proportion of persistent colonizers are able to consistently maintain relative abundances similar to those of the respective donors within the two HMA animal models. The HMA piglets, however, had higher proportions of persistent colonizers with donor-like relative abundances compared to the HMA mice.

**Phylogenetic analysis of Firmicutes colonizing the two HMA animal models**. As noted previously, ASVs of the phylum Firmicutes were the most conspicuous in terms of their differential establishment between the two HMA animal models. To further identify phylogenetic lineages that may affect colonization of core ASVs of the phylum Firmicutes among the two animal models, we analyzed the phylogenetic relationships among the core Firmicutes ASVs identified in the 4 donors (Fig. 6).

In total, we identified 178 core ASVs across the 4 donors which belonged to the phylum Firmicutes. Out of these 178 ASVs, 131 (74%) were identified as colonizers of the HMA piglets while only 69 (39%) colonized the HMA mice. Twenty-four of the 69 (39%) Firmicutes ASVs colonizing the mice were persistent colonizers while 74 out of the 131 (56%) were persistent colonizers of the piglets. As shown in Fig. 6, we identified 3 clades as preferentially colonizing the HMA pigs. This included a clade consisting of members of the family Christensenellaceae and two other clades consisting of members of the order Oscillospirales. We also observed five closely related Lachnospiraceae ASVs which persistently colonized both animal models. In addition, we identified a clade consisting of Clostridia UCG-014 that failed to colonize either animal model, with the exception of a single ASV which colonized the HMA piglets (Fig. 6).

**Core microbiota establishment in HMA animals using a different set of donors**. As described in Table 1, markedly low percentages of colonizers, as well as persistent colonizers were identified for HMA mice compared to HMA piglets for Donor_2, Donor_3, and Donor_4. In order to further verify these observations and investigate the potential donor-to-donor variation in colonization efficiency, we performed a second study where we inoculated mice with fecal inocula from a different set of donors (Donor_5, Donor_6, and Donor_7; see "Methods" for donor information) and provided the same solid diet. Core ASVs (i. e., present in all inoculum aliquots), "colonizers" and "persistent colonizers" were identified using the same approach. These donors were also selected to represent the variation in the human microbiome (Supplementary Fig. 5) and belonged to similar age groups as our first set of donors. In addition, since Donor_1, an infant donor, was the only donor from our first set of donors which had a bacterial community that established more successfully in HMA mice compared to HMA piglets, we investigated the repeatability of this observation by performing another inter-species transplantation experiment using fecal matter from a different infant donor (Donor_8) to GF mice (C3H/HeN) and GF piglets (*Sus scrofa domesticus*), similar to our first study.

The results of this subsequent study (Table 2) demonstrated that the Donor_7-HMA mice had comparable percentages of colonizers and persistent colonizers relative to Donor_4 HMA mice (Both Donor_4 and Donor_7 were seniors; see "Methods" for details). A similar comparison of core ASV establishment among the child donors (Donor_2 and Donor_5) revealed that these two groups of mice had very similar levels of persistent colonizers while Donor_5-HMA mice had a greater percentage of colonizers (Tables 1, 2). Donor_6-HMA mice had notably higher percentages of colonizers, as well as persistent colonizers, compared to the similarly-aged donor (Donor_3) of the first study (Table 2). These results demonstrate the possibility of having considerable donor-to-donor variation in colonization efficiency when transplanting human fecal bacterial communities into GF mice. The taxonomic distribution of the colonizing core ASVs in the HMA mice (Supplementary Data 10) was consistent with our previous observations – Bacteroidetes members colonized with high efficiency while Firmicutes ASVs established poorly (Supplementary Data 10). In agreement with our previous

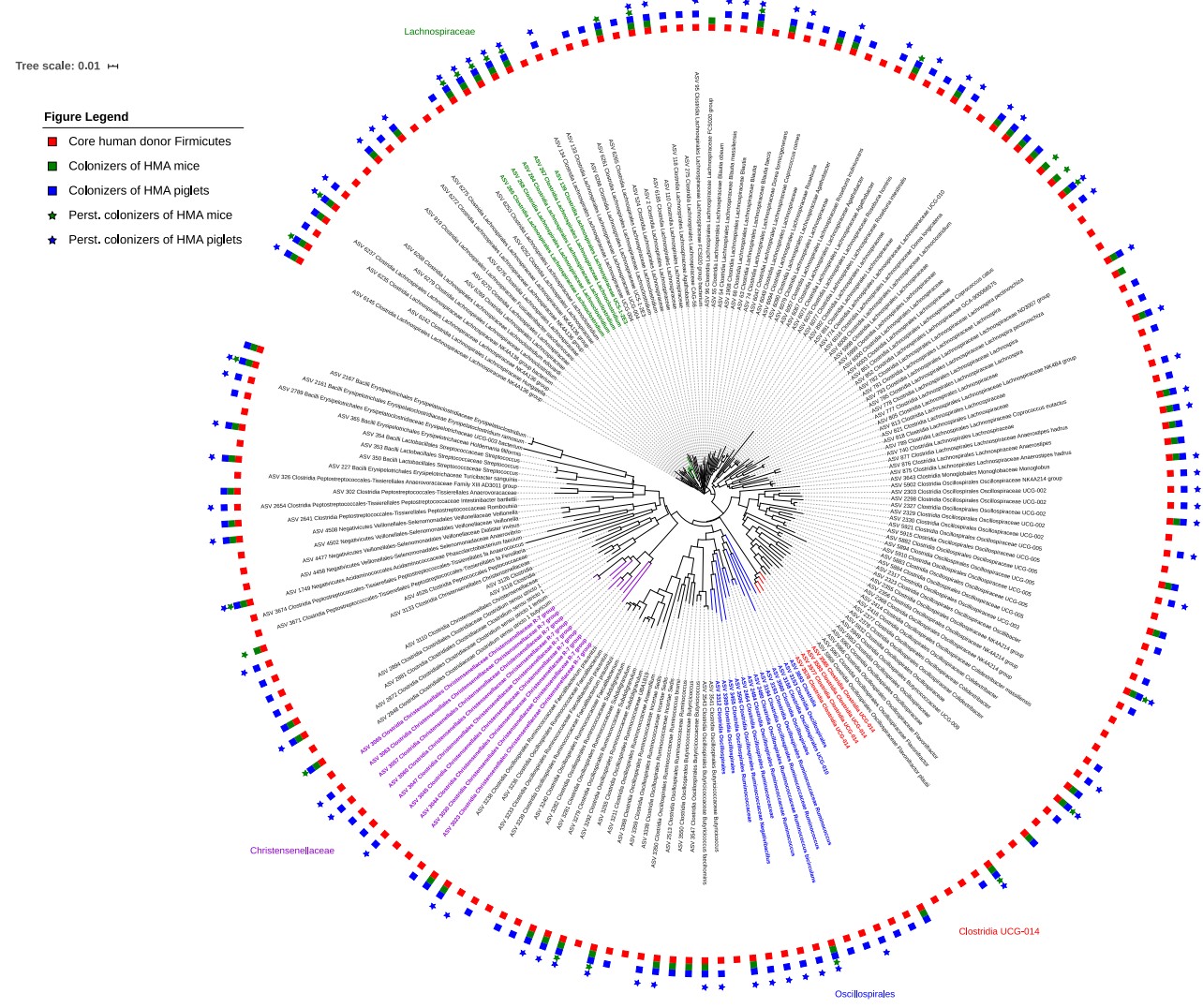

**Fig. 6 Phylogenetic tree depicting the overall distribution of core donor ASVs classified as Firmicutes among the two HMA animal models.** Red squares indicate the presence of core Firmicutes ASVs in the human donors while the green squares and blue squares indicate which of those Firmicutes ASVs were able to colonize the HMA mice and piglets, respectively. Green stars indicate persistent colonizers of HMA mice and blue stars indicate persistent colonizers of HMA piglets. $n = 13$ (3 piglets/donor, with the exception $n = 4$ for Donor_1, 2 days post inoculation) and $n = 37$ mice (Donor_1 = 7, Donor_2 = 10, Donor_3 = 10, Donor_4 = 10).

**Table 2 Summary of establishment of core donor ASVs in HMA animal models transplanted with a second set of donors.**

| Donor | Number of core donor ASVs | HMA animal model | Number of colonizers | Number of persistent colonizers |
|---|---|---|---|---|
| Donor_5 | 107 | Mice | 68 (63%) | 29 (27%) |
| | | NA | NA | NA |
| Donor_6 | 117 | Mice | 85 (73%) | 43 (37%) |
| | | NA | NA | NA |
| Donor_7 | 151 | Mice | 63 (45%) | 36 (24%) |
| | | NA | NA | NA |
| Donor_8 | 20 | Mice | 16 (80%) | 13 (65%) |
| | | Piglets | 18 (90%) | 11 (61%) |

*NA–not applicable.*

results, ASVs belonging to the phyla Lachnospiraceae and Ruminococcaceae displayed low colonization success, with the exception of the mice inoculated with Donor_6, which were efficiently colonized with Lachnospiraceae members (Supplementary Data 10). Additionally, the mice inoculated with Donor_6 and Donor_7 were poorly colonized by members of the family Oscillospiraceae. In contrast to the observation with the previous set of donors, some Christensenellaceae ASVs from Donor_6 (1 out of 1) and Donor_7 (1 out of 6) colonized the GF mice of this second study.

Similar to the results for our first infant donor (Donor_1), the core bacterial community of the donor inocula of our second infant donor (Donor_8) was more similar to the donor in the HMA mice than HMA piglets (Supplementary Fig. 6). Once again, the major taxonomic groups were established at much lower relative abundances in the Donor_8 HMA piglets compared to both the donor and the HMA mice (Supplementary Data 10).

**Comparison of predicted function between the human donors and the HMA animals.** To determine the potential functional consequences of the differential establishment of fecal bacterial communities between the two HMA animal models, we used PICRUSt[34] to predict the functional capacity of the human donors and the corresponding HMA animal models. Using the PICRUSt-predicted functional annotations, we identified the KEGG Orthology (KO) features that were detected in all the inoculum aliquots of a given donor and defined them as 'core predicted KO features' for each donor microbiome. Subsequently, this information was used to compare potential functions of the microbiomes established in the HMA models against the donors. The results, represented as PCoA plots (Supplementary Fig. 7), showed a similar clustering pattern to what we observed with the core ASVs (Fig. 2) where the Donor_1 inocula clustered closer to the HMA mouse samples while for the remaining donors, the donor inocula clustered closer to the HMA piglet samples. Thus, these results indicate that the differences in establishment of human donor microbiotas between the two animal models also results in potential differences in their functional capacities. As such, function may have favored the established taxa. Therefore, our belief is if similar taxa to the donor are established, similar function would also be present in the corresponding animal model.

## Discussion

To the best of our knowledge, this is the first study that directly compares HMA mice and HMA piglets to evaluate the establishment of fecal bacterial communities originating from the same human donors under similar dietary and environmental conditions. Using amplicon sequence variants, we compared the establishment of taxa from a total of five separate human donors in GF mouse and piglet models and an additional 3 donors in only the GF mouse model. We observed that the piglet model established a microbiota more similar to that of the human donors, with the exception of the two infant donors (i. e., Donor_1 and Donor_8), in which case the HMA mice established a more donor-like microbiota than the HMA piglets (Fig. 2 and Supplementary Fig. 6). Compared to the HMA mice, not only did the HMA piglets establish a considerably higher number of core donor ASVs but also harbored higher percentages of ASVs identified as persistent colonizers (Table 1). Notably, ASVs belonging to the phylum Firmicutes established more successfully in the HMA piglets compared to the mice (Fig. 6 and Supplementary Data 7). The phylogenetic analysis demonstrated certain groups of related Firmicutes ASVs to preferentially colonize the pig compared to the mouse. In addition, certain ASVs present in the donor did not colonize either model. Since the HMA mice were inoculated with smaller volumes of inocula compared to the inocula volumes used for the HMA piglets (see "Methods" section) it is possible that some of the lower abundance taxa in the donor inocula may have failed to transfer to the mice during gavaging. If this was indeed the case, then one would expect a pattern where the ASVs which did not colonize the mice would be low abundance members in the donor inocula, while the ASVs which successfully colonized the mice would be high abundance members. However, the data presented in Supplementary Data 1 show no evidence of such a pattern (i.e., there were both high abundance and low abundance core ASVs present in the donor inocula which did not colonize the mice) indicating that the lower inoculum volumes used to inoculate the mice could not have accounted for the low colonization numbers observed in these animals. Therefore, the differential colonization patterns observed for the two animal models may be a result of host adapted features present within these microbes and it would be of interest to

identify what factors help colonization. Future studies utilizing genome centric approaches focused on colonized and non-colonized isolates may shed light into features that help colonization. Such information would be critical to develop successful probiotics for microbiome manipulation aimed at improving human health. An analysis based on predicted metabolic functions also revealed a similar trend where the predicted functional capacity of the HMA piglets was more similar to that of the human donors with the exception of the infant donor, where HMA mice were functionally more similar to the human donor (Supplementary Fig. 7). Out of all the donors, the two infant donors had the lowest alpha-diversity (Fig. 1 and Supplementary Fig. 5). Thus, it is tempting to speculate that the bacterial diversity of the inoculum may influence colonization. However, the limited number of donors utilized in this study precludes us from making any robust conclusions in this regard. Future research with a wider range of donors will lead to a better understanding of how the microbiota diversity of the donor impacts the success of engraftment in animal models.

It is likely that host-related factors play a key role in favoring the establishment of human gut bacterial communities in a porcine host compared to a murine host. The domestic pig is widely used as an animal model in human infectious disease research[35], nutritional studies[36], and the study of cardiovascular diseases[37] due to the many anatomical, physiological, and immunological similarities[12,38] that pigs share with humans. The domestic pig is also considered the most human-like of all non-primate animal models[39]. Of particular importance to the gut microbiota is the higher level of morphological and functional similarities between the porcine gastrointestinal tract and the human gut compared to widely used rodent models[40,41]. Importantly, the porcine immune system is considered to be much more similar to the human immune system compared to that of mice[42,43], and the immune system plays a key role in structuring the microbiota during early development[44–46]. The size and volume of the gastrointestinal tract available for colonization might also be an important factor that differentiates the two animal models in terms of establishing human-like gut bacterial communities, especially in terms of the number of available niches for colonization by a complex microbiota. In humans, the gut epithelium is thought to play a role in selecting for gut microbes by stimulating the growth of certain microbial groups while suppressing others[47]. For example, host secretions such as fucosylated oligosaccharides are known to actively recruit Bifidobacteria and certain *Bacteroides* spp. in the infant gut lumen[47]. Although much less is known about how host factors in animals play a role in selecting for certain gut microorganisms, it is likely that similar mechanisms are at play. It can be speculated that, if more host related factors are shared between humans and pigs compared to mice, then this would favor the establishment of a more donor-like microbiota in the porcine gut compared to a murine gut. Thus, it is likely that anatomical, physiological, and immunological similarities of the donor and recipient may influence colonization of a more human-like microbiota in the recipient animal model.

For each donor, there were several core donor ASVs that failed to establish in either animal model (Supplementary Data 4). It is possible that some of these taxa might be human-adapted strains which do not establish in other hosts; however, this conclusion requires further experimental verification. These results underlie the importance of characterizing the establishment of taxa of interest in HMA animal models in human-to-animal microbiota transplantation experiments, as the human taxa suspected to be the cause of a human health outcome might not be able to colonize the animal model being used. In fact, a recent study evaluating >1700 samples from human-to-GF mouse transplant

studies in the published literature reported that, on average, <50% of species level taxa identified in the human donors able to establish in GF mice[48].

Compared to the number of persistent colonizers that colonized each animal model, only a small fraction of these ASVs established at relative abundances comparable to levels observed in the respective donor's consistently (Supplementary Data 9). This observation is likely related to the different ecological environments found in the guts of the two animal models compared to the human intestinal environment, as well as dietary differences between the animal models and humans. Providing diets that more closely resemble the diet of the human population under study to the HMA animal model may potentially aid in the establishment of a more human-like microbiota. Our analysis also revealed instances in which comparisons made at higher taxonomic levels masked notable differences which existed at the underlying ASV level. For example, for the Donor_3 core, the abundance of Lachnospiraceae was comparable between the human donor, HMA mice, and HMA piglets (mean abundances of 22.15%, 18.74%, and 25.83%, respectively; Supplementary Data 7). However, only 15 out of 40 core Lachnospiraceae ASVs colonized the HMA mice as opposed to the 31 out of 40 Lachnospiraceae ASVs which colonized the HMA piglets. Therefore, to accurately monitor establishment of the microbiota in surrogate models, it is critical to look at lower taxonomic levels to ensure colonization of representative species and their comparable abundances.

We observed considerable variation in the number of core donor ASVs identified in each donor and the success of their establishment in the two HMA animal models. Even in donors of similar ages, such as the two children (Donor_2 and Donor_5), we observed a marked difference in the number of core ASVs identified. This observation is in agreement with the well-known interpersonal variation in human microbiota composition[49]. We also noted considerable differences in colonization efficiency of core ASVs in the HMA mice even when inoculated with donors of similar age. This was most notable for the HMA mice inoculated with the two adult donors (Donor_3 and Donor_6; Table 1, 2). These results indicate that donor-to-donor variation in colonization efficiency is an important factor to consider in human-to-animal microbiota transplant studies. Thus, individual characteristics of a particular donor—such as genetics, dietary habits and lifestyle (which in turn affect the structure and composition of their microbiota)—may greatly influence the colonization success of their microbiota in HMA animal models.

We compared the transplantation results from our study with previous transplantation experiments performed using HMA mice and HMA pigs. The successful establishment of members of the phylum Bacteroidetes in both GF mice and piglets, as well as the enrichment of members of the genus Bacteroides in GF mice is in agreement with previous research[10,50–52]. In contrast, previous studies have reported that Bifidobacterium spp. tend to be very inefficient at colonizing GF mice[52,53]. However, in our study we noted that, across donors, almost all Bifidobacterium ASVs were able to colonize the HMA mice although mostly at lower abundances compared to the donor. These previous studies have used mice of different genetic backgrounds (e.g., C57BL/6[52]) compared to the C3H/HeN mice used in our study which might be a reason for this observation. The low success of colonization of Faecalibacterium spp. in the HMA mice of this study concurs with previous reports[52]. Concerning the establishment of Firmicutes, a study comparing the establishment of a human donor (32-year-old healthy adult) microbial community in different rat and mouse models noted inefficient establishment of Firmicutes in the mouse models[50]. Specifically, it was observed that eight abundant Clostridia phylotypes found in the human donor had

established in rats but were not detected in any of the mouse models, while a further six abundant phylotypes belonging to Clostridia failed to colonize either the rat or mouse models. In addition, only about 9 out of 48 Clostridia cluster IV members found in the human donor were detected in the mice. While the GF mouse strains used by these authors (C57BL/6JZtm and NMRI/MaxZtm) were different from the C3H/HeN mice used in our study, these results concur with our observation of poor establishment of Firmicutes in the HMA mice across donors. Thus, it appears that the inefficient colonization of GF mice with Firmicutes of human gut origin may be a common limitation across mouse strains which merits further investigation.

The limited number of published studies relating to the transplantation of human donor microbiotas into GF pigs hinders the ability to compare our results to the published literature. This is especially so for comparing the colonization patterns for our more mature donors as most of the studies related to HMA pigs involve transplantations from infant donors[7–9,11,14,15]. Even in one of the few studies which involved an adult donor, the results were reported only at the phylum level[16]. Therefore, there is a clear need for more research to be conducted with HMA pigs using human donors of diverse age groups and physiological conditions. For infants, a study using an HMA piglet model reported that although a majority of OTU (operational taxonomic unit)-level taxa found in the donor had colonized the piglets, their abundances had deviated from those of the donor upon transplantation[8]. This corresponds to the results obtained for both infant donors (Donor_1 and Donor_8) in our study where we observed that while a majority of the core ASVs colonized the HMA piglets, their abundances had deviated considerably from those observed in the donors (Supplementary Data 7 and 10). Finally, our observation of Bifidobacterium Spp. successfully colonizing the HMA piglets is in line with previous work[8,10].

The results of our study point to judicious selection of HMA animal models for future studies based on several model considerations. As an example, the murine model might be a better option for studying a low diversity microbiota such as the human infant microbiota while the porcine model may be a better option for studying humans with more complex microbial community compositions. When considering an animal model to study an aberrant microbiota implicated in a human disease condition, an HMA piglet or mouse model could be selected based on the composition of the aberrant microbiota and its probability of colonizing the animal model. For example, if the aberrant microbiota has a high proportion of members of the family Lachnospiraceae, then an HMA piglet model may be a more appropriate model as opposed to an HMA mouse model since our results demonstrate that Lachnospiraceae members colonize better in the HMA piglets.

Although we selected human donors to represent major variations in the human microbiome that may be driven by age, we only have a limited number of donors from each age group which precludes us from extending our findings to age-related colonization patterns. The lack of donor replication is mainly a result of the increased cost and logistical difficulties related to maintaining large numbers of murine and porcine animals under GF/gnotobiotic conditions. This is reflected by the fact that many studies performed using HMA mice or piglets have used only a single human donor[8,10,11,24,50,52]. Therefore, future studies with greater numbers of human donors representing different age groups will shed more light into the generalizability of our findings to a wider context. Our comparison of the HMA mice and HMA piglets also did not take into account the differences in developmental stage between the mice and the piglets. For example, although we sampled both species at the same time intervals post-inoculation, a 5-week old mouse would be at a more advanced stage in its

development life cycle compared to a 5-week old piglet. What consequences this has on the ability of human gut bacteria to colonize each animal model is an interesting question which merits further research. Notwithstanding these limitations, our findings provide new insight into the colonization dynamics of human gut bacteria in HMA murine and porcine models, which would be useful for future studies utilizing these animal models.

In conclusion, this study provides evidence for the utility of both HMA mice and HMA piglets as valuable animal models in human gut microbiota research depending on the donor context. The results of our study also emphasize the necessity to characterize the transplanted microbial communities at finer taxonomic levels (such as OTU or ASV level) when evaluating the success of donor microbiota engraftment in HMA animals, as comparisons done at higher taxonomic levels (e. g., Phylum, Family level) may lead to misleading conclusions. Our findings also point to the importance of confirming the establishment of targeted human taxa in the recipient animal models in human-to-animal transplantation experiments as several core human taxa from our donors failed to colonize either animal model.

## Methods

**Animals and husbandry practices**. All animal-related intervention protocols and husbandry practices were approved by the Institutional Animal Care and Use Committee (IACUC) at the University of Nebraska-Lincoln (protocols 1215 and 1700 for the mice and 1400 for the piglets). Male and female germ-free (GF) C3H/HeN mice were born and reared in flexible film isolators and maintained under gnotobiotic conditions at the University of Nebraska-Lincoln (UNL) gnotobiotic mouse facility. GF status of experimental mice was confirmed prior to study initiation as previously described[54]. At the beginning, there were 10 mice (5 male and 5 female) per isolator, with one isolator for each donor. However, 3 mice in the isolator inoculated with Donor_1 inocula died during the study. All mice in a given isolator were colonized with inocula from a single donor. Similarly, there were 10 mice per isolator at the beginning of the second study (involving Donor_5–Donor_8). One mouse each from the isolators inoculated with Donor_5 and Donor_6 inocula died during the study.

GF piglets were derived from two full-term (120-day) sows (Landrace × Duroc cross-bred) artificially inseminated with the same semen lot. The germ-free piglets were derived through hysterotomy via Cesarean section surgeries performed within the confines of pre-sterilized, custom-built plastic surgery isolators attached to the skin of anesthetized sows[55,56]. The newly-born piglets were placed inside sterile 'transfer boxes' while still inside the surgery bubble and transferred into sterile isolators with positive pressure (Park Bioservices LLC, MA). The isolators were tested for sterility using environmental swabs prior to introduction of the piglets. The swabs were cultured on brain heart infusion (BHI) agar plates and incubated both aerobically and anaerobically at 37 °C. Fecal swabs from the piglets were collected prior to inoculation with human donor inocula and were cultured similarly on BHI agar plates to test for germ-free status. Similar to the mice, the piglets (male and female) were divided among 4 isolators (one isolator per donor). There were 4 piglets in the isolator inoculated with Donor_1 and 3 piglets per isolator for the remaining 3 isolators. During the course of the experiment, one piglet from the isolator inoculated with Donor_1 died (fecal sample from 2 days post-inoculation was collected from this piglet prior to its death), resulting in a final total of 12 piglets. A single isolator with 4 piglets was colonized with Donor_8 inocula in the second study. All 4 piglets survived the duration of this study.

The mice were kept on Teklad 2919 (irradiated) rodent diet (Envigo RMS, inc., IN) throughout the study after weaning while the piglets were initially provided with commercially sterile human infant formula (Similac Pro-Advance®, Abbott Laboratories, IL) for 4 weeks and subsequently weaned on to the same Teklad 2919 rodent diet (This diet met the nutritional requirements of the piglets; see Supplementary Data 11 for diet composition). Sterile water was provided to both species in the form of autoclaved water. A 14 h light/10 h dark cycle was provided for both the piglets and mice. The HMA piglets were initially maintained at 35 °C for the first week after birth and then the temperature was reduced gradually to reach 25 °C which was subsequently maintained for the remainder of the study. The HMA mice were maintained at 20–24 °C throughout the study.

The gnotobiotic mouse facility and the gnotobiotic pig facility at UNL are both housed in the same building (UNL Life Sciences Annex) in very close proximity to each other.

**Human donor fecal sample collection and inocula preparation**. Human donors were recruited and consent was given as described on the IRB protocol approved by the Institutional Review Board of the University of Nebraska-Lincoln (Approval number 20170116853EP). Fecal samples were obtained from four human donors: Donor_1 (female, 20-week-old infant), Donor_2 (female, 5 years of age), Donor_3

(female, 25 years of age), and Donor_4 (female, 79 years of age). All donors were free of antibiotic use during the previous 6 months, while Donor_4 was on non-antibiotic medication for a respiratory illness. The donors of the second study were as follows: Donor_5 (male, 9 years of age), Donor_6 (male, 33 years of age), Donor_7 (female, 73 years of age), and Donor_8 (female, 6-month-old infant). From each donor, fecal samples were obtained on five consecutive days. Upon receiving the samples to the laboratory, samples were mixed with a solution of 50% glycerol and stored at −80 °C. Prior to inocula preparation, all 5 fecal samples from a donor were thawed at 4 °C and transferred into an anaerobic chamber. Within the anaerobic chamber, 0.5 g from each fecal sample was weighed and pooled to obtain a composite fecal sample of 2.5 g. This composite fecal sample was subsequently mixed with 50 mL of sterile Similac® infant formula and thoroughly mixed to obtain the master pooled inoculum. Aliquots of this pooled inoculum (referred to as 'inoculum aliquots') were subsequently used for inoculating the GF mice, as well as the GF piglets. Individual tubes with inoculum aliquots were stored at −80 °C until required to perform the inoculations. The tubes contained sterile 20% glycerol to ensure the inoculum did not completely freeze.

**Inoculation of germ-free mice and piglets**. Each mouse was inoculated with 200 µL of donor inoculum aliquots while the piglets were inoculated with 4 mL of inoculum aliquots per piglet (this volume was selected based on the body weight ratio between the mouse and the young piglet). The mice were orally gavaged with the inoculum aliquots while for the piglets the inoculum aliquots were added to the feed bowls to be taken up during feeding. Both the mice and the piglets were inoculated after weaning and adaptation to the solid diet. The GF mice were weaned at 21 days of age and inoculated at 33 days of age. The GF piglets were weaned at 28 days of age and inoculated at 31 days of age. All animals in a given isolator were inoculated with the same donor. Following the first round of inoculations, all animals were inoculated a second time in a similar manner two weeks later. The remaining material from the inoculum aliquots were used for DNA extraction and subsequent sequencing to characterize the fecal bacterial community of each donor that went into each recipient GF animal.

**Fecal sample collection and storage**. From both animal models, fecal samples were collected at the following time points post-inoculation (number of days after first round of inoculations): 2d, 7d, 14d, 21d, 28d, 35d, and 40d. Subsequently, all animals were humanely euthanized. Fecal samples were collected using sterile fecal loops for the piglets. Fecal pellets from mice were collected by allowing each mouse to defecate into a plastic cup placed inside the isolator. All fecal samples were stored at −80 °C until they were used for DNA extractions.

**Fecal DNA extraction, PCR amplification, and amplicon sequencing**. DNA was extracted from all fecal samples using the OMEGA Mag-Bind® Soil DNA 96 kit (Omega Bio-tek, Inc., GA) following the manufacturer's protocol with the following modification: after the initial bead-beating step, the samples were boiled for 10 mins at 90 °C in a heated water-bath, followed by another round of bead-beating at 30 Hz for 10 mins. Samples were processed in batches of 96 and each batch contained mouse fecal samples, as well as piglet fecal samples in order to avoid any biases that may be introduced due to possible variations during DNA extractions. Following DNA extractions, PCR amplifications were performed using primers targeting the V4 hypervariable region of the 16S rRNA gene as previously described[57]. Each PCR reaction consisted of 1× Terra™ PCR Direct Buffer (Takara Bio Inc., Mountain View, CA), 0.625 units of Terra™ PCR Direct Polymerase (Takara Bio, Inc., Mountain View, CA), 2.5 µM barcoded primers[57], 20–50 ng of template DNA, and 9 µL of nuclease-free water for a total volume of 25 µL. PCR reactions were performed on an Applied Biosystems Veriti 96-well Thermal Cycler (Thermo Fisher Scientific, Inc., Waltham, MA) using the following program: Initial denaturation at 98 °C for 3 min followed by 25 cycles of denaturing at 98 °C for 30 s, annealing at 55 °C for 30 s, and elongating at 68 °C for 45 s. This was followed by a final extension at 68 °C for 4 min. Agarose gel electrophoresis at 2% (w/v) agarose (Green BioResearch LLC, Baton Rouge, LA) was used to visualize the resulting amplicons. Following amplicon normalization using the SequalPrep Normalization Plate Kit (Invitrogen, Carlsbad, CA), equal volumes of normalized amplicons were pooled together. Preparation of PCR plates, amplicon normalization, and pooling were performed using an EpMotion M5073 robot (Eppendorf, Hamburg, Germany). Subsequently, the quality of the pooled amplicons was assessed using an Agilent BioAnalyzer 2100 High Sensitivity DNA chip (Agilent, Santa Clara, CA) and quantified using a DeNovix dsDNA High Sensitivity Kit (DeNovix Inc., Wilmington, DE). Dual-index paired-end sequencing (2 × 250 bp) of pooled amplicons was performed on an Illumina® MiSeq™ platform (Illumina Inc., San Diego, CA) as described previously[57] using V3 reagent kits.

**Bioinformatic analysis**. The bioinformatic data processing and analysis steps were performed in R[58] (version 3.6.0) using the phyloseq package[59] (version 1.28.0). Briefly, denoising steps and identification of amplicon sequence variants (ASVs) were performed following the DADA2 (1.12.1) pipeline[60]. Potential contaminant ASVs originating from reagents were identified and removed using the decontam package[61] (1.4.0) based on their prevalence in negative controls. Following chimera removal, DNA extraction and PCR negative controls were removed along with

ASVs found only in those negative controls. Taxonomic assignments were done using the DADA2-formatted training fasta file[62] from the SILVA[63] Project's version 138 release. ASVs classified as 'Archaea', 'Eukaryota', and 'Cyanobacteria' (with the exception of the non-photosynthetic class Melainabacteria, as these are thought to be of gut origin[64]), as well as ASVs which were only detected in a single sample were also filtered out. Any sample with less than 10,500 reads was also removed from the analysis. The final quality-filtered ASV table consisted of 392 samples and 1135 ASVs accounting for 17,016,489 total reads at an average read depth of 43,409 reads/sample (SD = 21,178) which was used in the subsequent data analysis steps. Beta diversity analysis was done using unweighted UniFrac and Bray-Curtis distance matrices and alpha diversity comparisons were performed using the Shannon Index. All ASVs (i.e., core and non-core ASVs) were considered for alpha diversity calculations as these estimates are sensitive to the presence/absence of rare taxa. Heatmaps and chord diagrams were generated using the R packages 'gplots'[65] (3.0.4) and 'circlize'[66] (0.4.10), respectively. Metagenome prediction based on 16S rRNA gene sequences was performed using PICRUSt2[34] and the associated tools HMMER (http://hmmer.org), EPA-NG[67], gappa[68], and castor[69]. The Firmicutes phylogenetic tree (Fig. 6) was generated using the Interactive Tree of Life (iTOL)[70] web interface.

In addition, the taxonomic assignments of ASVs identified as *Faecalibacterium prausnitzii* (ASV_3233, ASV_3238, and ASV_3239), *Bifidobacterium bifidum* (ASV_3778 and ASV_3779), and *Ruminococcus bromii* (ASV_3350) were verified with the Basic Local Alignment Search Tool (BLAST)[71] against the nucleotide collection (nr/nt) database using megablast (optimized for highly similar sequences).

**Statistics and reproducibility**. For comparing the relative abundances of persistent colonizer ASVs in the HMA animals with their abundances in the original donor inocula at each sampling time point, DESeq2[72] (1.24.0) was used as implemented through phyloseq. p-value adjustments for multiple comparisons was performed using the Benjamini–Hochberg procedure[73]. Alpha diversity measures (Shannon index), as well as UniFrac and Bray–Curtis distances (Fig. 3) were compared using the Wilcoxon rank-sum test[74]. Statistical significance in all analyses was determined at $p < 0.05$.

No sample size determination was done prior to the start of the experiments since the number of animals that can be maintained within gnotobiotic isolators is limited by logistical and animal welfare considerations (This was especially the case with the HMA piglets).

**Reporting summary**. Further information on research design is available in the Nature Research Reporting Summary linked to this article.

## Data availability

The raw sequence data generated and analyzed during this study are available under BioProject accession number PRJNA560867 from the National Center for Biotechnology Information (NCBI) sequence read archive (SRA). Source data for generating all main figures are provided in Supplementary Data 12. All other relevant data are available from the authors upon request.

## Code availability

All code related to the bioinformatic analysis of this study (including reproduction of all figures) are documented as an R markdown file and is available in the GitHub page of the Fernando Lab (https://github.com/FernandoLab/men_mice_and_pigs) along with all the related metadata and accessory files.

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

## Acknowledgements

We wish to express our sincere gratitude to the staff members of the UNL Gnotobiotic mouse facility, as well as Bayliegh Bohn and other staff members of the UNL Life Sciences Annex for their technical expertize and help with animal handling and care. We would also like to thank Jeff Perkins at the Eastern Nebraska Research and Extension Center (ENREC) for management of the sows used in this study. A.R.T. and S.C.F. acknowledge support received from the Nebraska Food for Health Center (NFHC) of the University of Nebraska-Lincoln.

## Author contributions

S.C.F., N.D.A., T.E.B., and A.R.T. designed the study. A.C.B., W.A.T., and N.D.A., performed the study and were responsible for animal husbandry aspects related to the HMA piglets. H.K., R.J.S., and N.D.A. performed the study and were responsible for the animal husbandry aspects related to the HMA mice. K.D.H. and C.K.-B. performed the C-section surgeries to derive the GF piglets. P.S.M. oversaw the artificial insemination and the management of the sows used for deriving the GF piglets. N.D.A. performed the bioinformatic analysis. N.D.A. and S.C.F. wrote the manuscript which was subsequently revised by A.R.T., T.E.B., W.A.T., A.C.B., H.K., K.D.H., and C.K.-B.

## Competing interests

The authors declare the following competing interests: S.C.F., an author of this publication has disclosed a significant financial interest in NuGUT LLC. In accordance with its Conflict of Interest policy, the University of Nebraska-Lincoln's Conflict of Interest in Research Committee has determined that this must be disclosed. The remaining authors declare no competing interests.
