## [Peer Review File · Communications Biology]

Reviewers' Comments:

Reviewer #1:

Remarks to the Author:

I find the subject of this paper very relevant. Often papers dealing with FMT report effects or no effects in the recipients but to my knowledge few studies report the level of engraftment in detail. This is a very important parameter because for example lack of effect in the recipient could be due to lack of engraftment but also to lack of effect in the recipient due to individual differences, the donor and recipient are different species (as in the present study),...

The issue of very few donors is a drawback here (and in many other studies) to make too conclusive statements on differences in engraftment.

As reflected in my comments, the design of the study should be described with more clarity and more details both in order to interpret the results and in case of wish to reproduce the work by others.

This is an interesting manuscript dealing with a relevant subject that is often not covered or poorly described in this type of studies practicing FMT, i.e., engraftment. Therefore, the manuscript contributes with new information.

The conclusions regarding how age can influence engraftment should not be too strong since only 1 donor per age was used in the main study and one more in a follow up study in mice. The differences among donors could be as much be due to individual variations and not necessarily age. This issue regarding too few donors (and also often few recipients) is a general weakness in this type of studies (it is expensive laborious....). SO the more studies published the more data will be available to draw more clear conclusions.

The paper could be considered for publication in Communications Biology but a number of aspects should be considered and modified before.

General comments:

- Add a little more concrete information at the end of the abstract (see my comment under specific comments too).

- The experimental design is not clearly described, number of animals per donor, number of isolators per donor, both in the first study and in the follow up study.... Also, some parts are described in the results, some in the materials and methods and some information is even described in the discussion (ex. Line 488). Another example: Line 620 you talk about repeat experiment. What is that? This has not been described. Why did you use different donors? You explain it in line 202 but this should be described in the materials and methods too, otherwise it is very unclear.

Also from line 87 to 95 is in fact materials and methods not results.

Please improve the description of your design throughout the text, it will help the reader very much.

- The weakness of having only one donor per age category makes conclusions on effect on age on engraftment difficult. You might have found similar differences if you had used four different donors of the same age category but from different origins or dietary habits,... You mention it at the end of the discussion but in my opinion it should be moved to the beginning of the discussion so that you tell the reader that you are aware of it already before you start discussing the results.

- In your discussion, I miss that you compare your results with the literature. There are not many

publications reporting engraftment, but there are some. Please compare your results with the published results.

- The graphs are hard to read. Should be of better quality.
- Please add the number of replicates to all fig legends.

Specific comments (the same comments are in the pdf file as comments attached to the specific lines. This is only intended as a help, that is, you do not need to look at the comments in the manuscript since they are the same as those here):

-Line 36-38: Please make this statement more precise by shortly describing what this refers to?

-Line 51: In the next paragraph you say that to move beyond correlations, HMA are being conducted, but this reference is already on HMA. So please use other references to make your point here, which according to your discussion should not be dealing with HMA, which will be the next step.

-Line 93-94: how many days after weaning? When were they weaned?

-Line 95: Could you please show what the diet composition was (to see how much it deviates from a standard pig diet)

-Line 99-101: 4 donor inocula from same source????? Unclear.

Do you mean 4 samples from the same donor? So, please write it this way instead. Please, distinguish here and elsewhere between inoculum (the pooled material you use for transplantation) and samples, which are the 5 faecal samples taken from each individual. You have taken 5 samples per donor, why is it reduced to 4 and 3? Were they discarded due to too low sequence number? If so, please describe. If not describe the reason for this lower number of samples.

-Line 105: So for ex. for the infant, 26 ASV accounted for 99% of the reads??? And for the adult 140 for 98% of the reads??

Sounds low, is this low or in agreement with what is found in similar samples in other studies?

-Line 108-109: What do you mean by all inocula of a given donor? you only make 1 pooled sample that you inoculate right?

So you mean in all donor samples? Please clarify.

-Line 110-111: As far as I understand, you have used the core ASVs in all except diversity results, I think you should state it here to make it clear for the reader (although I know you describe it in the M&M)

-Line 113: So in this section you compare the whole microbiota of the recipients with the core of the donors? Please make it clear.

-Line 140-141: Looking at the graph, this is not completely true for all cases (infant and adult some as low as day 2). Please rephrase.
Also are differences significant??

-Line 144: In this section, I suggest to reduce the text so that data already presented in table 1 and 2 are not repeated again in the text.

-Line 157: You had 7 sampling time points, so you mean in at least 4 out of the 7 time points? (As you do for the donor like abundances in line 398? If so, please modify write 4 out of 7 instead to

make descriptions more comparable.

-Table 1: Please, remove the decimal in the % values. Not necessary in my opinion, and will make the table easier to read.

The same for Table 2.

-Line 167-170: Delete. Already in Table.

-Line 177-179: Delete. Already in Table.

-Line 181-185: Delete. Already in Table.

-Line 187-189: Delete. Already in Table.

-Line 191-192: Delete. Already in Table.

-Line 195-196: Delete. Already in Table.

-Line 207: Also a big difference in the number of core donor ASVs!!! Please add to the discussion.

-Line 211-213: This means that if you had repeated the study not only with mice but also with pigs, perhaps the data on pigs had been lower and the conclusion would be the opposite! i.e., that a lower engraftment had been seen in pigs than in mice. This shows how vulnerable it is to test only one donor. Please add this to the discussion.

-Line 218-220: I suggest to delete the decimal in the % values shown in this section.

-Line 323-324: The infant has a lower number of ASV but also abundance than the other donors. Was the infant breast-fed? If yes, isn't this an unexpected result? Please add in discussion.

-Line 396-398: In how many animals? Please add.

-Wouldn't it be logical to say in at least 50% of the animals as you say for persistent colonizers?

So it would mean that all ASVs classified as 'donor like in abundance' are also persistent colonizers but not all persistent colonizers are 'donor like in abundance'.

If not, I think it gets unnecessarily complicated and perhaps not so relevant if you consider also those ASV present in less than 50% of the animals.

But please let me know if there is a point behind doing it differently or if I have misunderstood something.

-Line 444-446: Following my comment above: wouldn't it be more relevant to say how many of the persistent maintain the abundance of the donor?

-Line 484: delete ,not markedly different' and write 'was similar'

-Line 488-489: This information on number of recipients should also be added to the materials and methods.

-Line 493-501: I find this too speculative taking into account the few donors used. If you had donors of same category and different diversities perhaps you could do this kind of correlations.

-Line 502: There is some discussion in the literature of this kind:
'It has been shown that the infant gut actively recruits Bifidobacteria and Bacteroides thetaiotaomicrons by secretion of fucosylated oligosaccharides (FOS) into their gut lumen. (In: ' Symbiosis, dysbiosis, and rebiosis—The value of metaproteomics in human microbiome monitoring. Lei Mao and Jacqueline Franke).
In my opinion, this type of discussion should also be included here when talking about engraftment.

-Line 520-523: This could be so but the fact that you only have one donor per age category makes this kind of argumentations difficult. If you had had 4 different donors of the same age but used to eat different diets, you could perhaps (probably) have found similar differences among the different donors. So, maybe it is more the individual variation (due to the many factors characteristics of each individual) than the age as a crucial factor.
This 'weakness' of the design should be more clearly emphasized.

-Line 559-564: This refers to my comment above and I suggest that you move this text to the beginning of the discussion so that you tell the readers already there that you are aware of the limitations of the setup.

-Line 572-575: How was the performance of the pigs (and the mice)? Feed intake, body weight? Please add a couple of sentences in the beginning of the results reporting these data to indicate whether the animals thrived or deviated too much from 'normal' physiological conditions.

-Line 584: I suggest to add here or in the abstract your point on line 537 and onwards regarding the level at which the data are looked at. That is, maybe similarities are found at for ex. family level but it looks very different at a lower level.

-Line 591-594: how were the mice housed?
how many in each isolator?
how many isolators per donor?
how many isolators in total?

-Line 601-602: How many isolators and piglets in total per donor and in total?
Is it 1 isolator per donor? That is, 4 isolators in total per animal species?

-Line 620-621: This is not clear. You should describe this procedure in more detail, explaining that you did a follow up study to look at variation among donorsWhat is the difference between the repeat study and the follow up study? This is unclear. I think you only had one follow up study where you used all 4 donors didn't you?
Please describe more clearly your procedure.

-Line 639-640: when were the animals weaned and how many days after weaning were they inoculated?

-Line 649: After second transplantation?

-Line 696-698: By 'this analysis' you mean alpha diversity only? Please make it clear since you mention both alpha and beta diversity in the previous sentence.

Reviewer #2:

Remarks to the Author:

Aluthge et al., described the impact of human microbiome on pig or mouse models to understand

the microbiome implantation. As in most of the study are performed in mice models, thus authors took the advantage of the piglet model to explore whether it can be use for human nutritional and biomedical studies as many of anatomical, physiological and immunological similarity between human and mice. This unique study explored the direct engraftment of fecal bacterial from human donors of varying age range (infant, child, adult and elderly). Using 16S rRNA gene analysis using amplified sequence variants identified that more engraftment in piglet compared to the mice for child, adult and senior donors. But mice model is better off for the infant donor taxa. These results obtained in this study are novel and certainly interest to others in the community and the wider field.

I have some major and minor comments which could be helpful to improve the manuscript.

Major comments

1. Main concern of this study that authors used 16S rRNA sequence which does not allow to identify the functions of microbiome. Perhaps, authors should show the functional study using shotgun sequencing.
2. Both mice and piglet have different immune system may be due to these authors found the engraftment variability for child, adults and seniors rather than the size of the gut in the piglet which could accommodate more bacterial load.
3. The genera Bacteroides and Bifidobacterium colonize both animal models- is there functional similarity with the host genetics that's why both taxa are selected.
4. Authors should another GF WT model (C57BL/6) to identify the genetics of mice does not affect the microbiome implantation.

Minor comments

1. Fig 1 and 3, poor quality of figures.
2. Do piglets have any cage effects?

Responses to Reviewers

We thank the reviewers for their constructive suggestions. Below we give detailed explanations and have updated the manuscript with responses based on reviewer comments. All corrections in the revised manuscript is highlighted to help the reviewers track changes.

Reviewer 1

General comments:

- Add a little more concrete information at the end of the abstract (see my comment under specific comments too).

This information has been added. Please see lines 29-38 of the abstract.

- The experimental designed is not clearly described, number of animals per donor, number of isolators per donor, both in the first study and in the follow up study..... Also, some parts are described in the results, some in the materials and methods and some information is even described in the discussion (ex. Line 488). Another example: Line 620 you talk about repeat experiment. What is that? This has not been described. Why did you use different donors? You explain it in line 202 but this should be described in the materials and methods too, otherwise it is very unclear.

The shortcomings described above have been addressed and detailed in the responses to the specific comments.

- Also from line 87 to 95 is in fact materials and methods not results

We agree with the reviewer that this information is related to materials and methods. However, this information was added deliberately to help the reader understand the experiments conducted before describing results as the “Results” precede the “Methods”. Therefore we believe this short description would help the reader interpret the data better. (now lines 89 to 99)

- Please improve the description of your design throughout the text, it will help the reader very much.

As suggested by the reviewer, we have revised the manuscript greatly describing the study design in the “Methods” section. Specific details are provided as responses to specific comments below.

- The weakness of having only one donor per age category makes conclusions on effect on age on engraftment difficult. You might have found similar differences if you had used four different donors of the same age category but from different origins or dietary habits,... You mention it at the end of the discussion but in my opinion it should be moved to the beginning of the discussion so that you tell the reader that you are aware of it already before you start discussing the results.

We recognize the shortfall of having only one donor for each age group. As a result (as suggested by reviewer 1 we have refrained from emphasizing the age factor in the manuscript and used the 4 donors to represent the variation in the microbiome. Additionally, to date no study has investigated engraftment of the human microbiome in both germfree pig and mouse models

using the same donor, as such we believe this study provides valuable information to the field. Combined with the donors of our follow-up studies described in the manuscript, this have result in a total of 8 individual donors. This would increase replication and make conclusions more robust and help overcome the age factor.

Finally, we have described the possibility of dietary variation among donors in the discussion (lines 610 to 611).

- In your discussion, I miss that you compare your results with the literature. There are not many publications reporting engraftment, but there are some. Please compare your results with the published results.

This has been added to the discussion (lines 614-653).

- The graphs are hard to read. Should be of better quality.

- Please add the number of replicates to all fig legends.

The quality of the graphs have been improved and the number of replicates have also been added to the figure legends. The figures are uploaded as separate files with improved quality.

Specific comments (the same comments are in the pdf file as comments attached to the specific lines. This is only intended as a help, that is, you do not need to look at the comments in the manuscript since they are the same as those here):

-Line 36-38: Please make this statement more precise by shortly describing what this refers to?
The abstract has been modified to include these details (lines 29-38).

-Line 51: In the next paragraph you say that to move beyond correlations, HMA are being conducted, but this reference is already on HMA. So please use other references to make your point here, which according to your discussion should not be dealing with HMA, which will be the next step.

We've changed the reference to 'Moving microbiota research toward establishing causal associations that represent viable targets for effective public health interventions' by Volker Mai et al to make our point about the necessity for the field to move towards establishing causality of the microbiome in human health outcomes.

-Line 93-94: how many days after weaning? When were they weaned?

The GF mice were weaned at 21 days of age and inoculated at 33 days of age. The GF piglets were weaned at 28 days of age and inoculated at 31 days of age. This information has been added to the 'Methods' section (lines 766-768).

-Line 95: Could you please show what the diet composition was (to see how much it deviates from a standard pig diet)

This information has been included as Supplementary File 11.

-Line 99-101: 4 donor inocula from same source????? Unclear.

Do you mean 4 samples from the same donor? So, please write it this way instead. Please,

distinguish here and elsewhere between inoculum (the pooled material you use for transplantation) and samples, which are the 5 faecal samples taken from each individual. You have taken 5 samples per donor, why is it reduced to 4 and 3? Were they discarded due to too low sequence number? If so, please describe. If not describe the reason for this lower number of samples.

It appears that our description of methods used to generate inoculums has caused some confusion for which we apologize. The 4 donor inocula sequenced for each donor refers to the 4 aliquots (2x for inoculating the mice and 2x for inoculating the piglets). These inocula were generated for each donor by pooling fecal material collected on 5 consecutive days. For example, we homogenized the 5 fecal samples provided by the ADLT donor and made the pooled material. Subsequently, aliquots were made from this same homogenized sample for all inoculations of that donor to both pig and mouse models for both the initial and reinoculation. These aliquots are described as “inocula” in the manuscript.

All aliquots were prepared at the same time and stored at -80 C until required for inoculation. To avoid confusion between the pooled material and the aliquots derived from the pooled material actually used to inoculate the animals, we have changed the term ‘inocula’ to ‘inoculum aliquots’ in the revised manuscript.

The reason for the INF donor having only 3 inocula was because we were unsuccessful in extracting good quality DNA from the second aliquot used to inoculate the piglets. In order to capture as closely as possible the bacterial community being introduced to the animals, we used what was remaining from each aliquot after inoculating the animals as our material for DNA extraction and subsequent sequencing which was then taken to represent the community of the donor inoculum at the time of colonization. The manuscript has been updated with these details in this section (lines 104-112) as well as in the ‘Methods’ section (lines 747-755).

-Line 105: So for ex. for the infant, 26 ASV accounted for 99% of the reads??? And for the adult 140 for 98% of the reads??

Sounds low, is this low or in agreement with what is found in similar samples in other studies?

As the reviewer has noted, these numbers do indeed seem low at first glance. However, we’re not considering every ASV which was detected in the donor inocula but only those ASVs which were detected in all 4 donor aliquots.

The main reason we wanted to look at a core microbiota was to account for variation in bacterial community composition that results from variations in sequencing depth that may influence rare ASVs identified. By concentrating on ASVs found in all inocula aliquots, we’re focusing on the major community members which do not fluctuate/vary as a result of sequencing depth and are consistently found in the donor. As these ASVs represented a major portion of the reads, this suggests these ASVs represent the major portion of the microbiome. We also believe that this approach of focusing on the shared ASVs help us filter out any ‘spurious’ ASVs that may be a result of sequencing errors.

Most studies in the literature that describe the total number of taxa identified across multiple human subjects do not concentrate on a core microbiota using the same criteria that we’ve used in this study, making it difficult to directly compare the number of ASVs detected in our donors with those reported in the literature. However, a recent publication (‘Baseline human gut microbiota profile in healthy people and standard reporting template’ by King et al., PLOS ONE publication, 2019) reported 155 total bacterial species across 50 healthy volunteers ranging in age from 19 – 39 years. These authors also identified a ‘core’ of 84 bacterial species that were

found in all subjects. Our ADLT donor (25 years) had 140 core ASVs. Kumar et al. in their study 'Impact of nutrition and rotavirus infection on the infant gut microbiota in a humanized pig model' (BMC Gastroenterology (2018)) seem to have detected 45 OTUs (based on Fig.1) from a single donor sample of their infant donor. Our core for the INF donor consisted of 26 ASVs shared across 3 inoculum aliquots.

-Line 108-109: What do you mean by all inocula of a given donor? you only make 1 pooled sample that you inoculate right?

So you mean in all donor samples? Please clarify.

Yes. One pooled sample was used for inoculation. This has been clarified above. We have changed the term to 'inoculum aliquots' to avoid confusion.

-Line 110-111: As far as I understand, you have used the core ASVs in all except diversity results, I think you should state it here to make it clear for the reader (although I know you describe it in the M&M)

This information has been added as suggested (lines 137-139).

-Line 113: So in this section you compare the whole microbiota of the recipients with the core of the donors? Please make it clear.

This section has been re-worded to make this clearer (lines 124-125).

-Line 140-141: Looking at the graph, this is not completely true for all cases (infant and adult some as low as day 2). Please rephrase.

Also are differences significant?? Upon performing the statistical comparisons between time points as suggested by the reviewer, we realized that our statement of the 2-day time point having the lowest alpha diversity only held true for the CHLD-HMA mice and SNR-HMA mice in terms of statistical significance. Thus, our statement is not generalizable across all donors. Therefore, we have removed this statement.

-Line 144: In this section, I suggest to reduce the text so that data already presented in table 1 and 2 are not repeated again in the text.

We have removed information already present in the tables as indicated by the reviewer.

-Line 157: You had 7 sampling time points, so you mean in at least 4 out of the 7 time points? (As you do for the donor like abundances in line 398? If so, please modify write 4 out of 7 instead to make descriptions more comparable.

The time points have been changed to '4 out of 7' as suggested (line 168).

-Table 1: Please, remove the decimal in the % values. Not necessary in my opinion, and will make the table easier to read.

The same for Table 2.

The decimal points have been removed from the % values as suggested.

-Line 167-170: Delete. Already in Table.

Deleted as suggested.

-Line 177-179: Delete. Already in Table.
Deleted as suggested.

-Line 181-185: Delete. Already in Table.
Deleted as suggested.

-Line 187-189: Delete. Already in Table.
Deleted as suggested.

-Line 191-192: Delete. Already in Table.
Deleted as suggested.

-Line 195-196: Delete. Already in Table.
Deleted as suggested.

-Line 207: Also a big difference in the number of core donor ASVs!!! Please add to the discussion.

While revisiting the code used for generating this data, we realized that we had not been looking at a core. We have re-run the analysis looking at core donor ASVs and table 2 has been updated accordingly. We've also discussed the differences in numbers of core ASVs in our discussion (lines 600-613).

-Line 211-213: This means that if you had repeated the study not only with mice but also with pigs, perhaps the data on pigs had been lower and the conclusion would be the opposite! i.e., that a lower engraftment had been seen in pigs than in mice. This shows how vulnerable it is to test only one donor. Please add this to the discussion.

We have revised our manuscript such that we don't emphasize the age factor as much as we did in the previous version. However, we have modified the paragraph which addresses the difficulty of drawing any conclusions regarding the influence of age in determining colonization efficiency due to the limited number of donors involved (lines 665-668). Interestingly, even in the donors of our follow-up study, we were once again able to observe the inefficiency of Firmicutes in colonizing the GF mice (Supplementary File 10). Taking into consideration that our results consistently show that the Firmicutes phylum colonize the pigs better than the mice and that the mice and pigs are comparable in their ability to be colonized by Bacteroidetes (These two are the major phyla of the human microbiome). As such we have updated the manuscript accordingly.

-Line 218-220: I suggest to delete the decimal in the % values shown in this section. The decimal values have been removed in all except when dealing with abundances and standard deviations.

-Line 323-324: The infant has a lower number of ASV but also abundance than the other donors. Was the infant breast-fed? If yes, isn't this an unexpected result? Please add in discussion. The dietary intake sheets provided by the mother of this infant indicate that the infant's primary diet was breast milk but was also supplemented with infant formula and even some solid food in

the form of germ oatmeal. Since the infant was not exclusively breastfed and also because there was some solid food in the diet, the *Bifidobacterium* numbers and abundances might have been lower than expected. Moreover, it is also known that a certain proportion of infants have very low levels of Bifidobacteria in their stool regardless of whether they were fed breast milk or formula (Tannock et al., 'Why Don't All Infants Have Bifidobacteria in Their Stool?', *Front. Microbiol.*, 2016). Therefore, we don't consider this to be an unexpected result.

-Line 396-398: In how many animals? Please add.

Since we have now focused on persistent colonizers (See response below), we are only identifying ASVs present in 50% of the animals for each species.

-Wouldn't it be logical to say in at least 50% of the animals as you say for persistent colonizers? So it would mean that all ASVs classified as 'donor like in abundance' are also persistent colonizers but not all persistent colonizers are 'donor like in abundance'.

If not, I think it gets unnecessarily complicated and perhaps not so relevant if you consider also those ASV present in less than 50% of the animals.

But please let me know if there is a point behind doing it differently or if I have misunderstood something.

-Line 444-446: Following my comment above: wouldn't it be more relevant to say how many of the persistent maintain the abundance of the donor?

Upon further reflection, we agree with the reviewer that it would be more relevant (and cause less confusion) if we concentrated solely on which of the 'persistent colonizers' were also able to maintain donor-like abundances. Therefore, this section has been updated with this information (lines 391-429).

-Line 484: delete 'not markedly different' and write 'was similar'

We have revised this section of the manuscript as suggested by the reviewer.

-Line 488-489: This information on number of recipients should also be added to the materials and methods.

This information has been added to the 'Methods' section (lines 701-707 and lines 719-724).

-Line 493-501: I find this too speculative taking into account the few donors used. If you had donors of same category and different diversities perhaps you could do this kind of correlations. In the revised manuscript we have modified this section and limited ourselves to stating that the microbiome diversity of the donor might be an important factor which influences colonization success in HMA animals, but that further research with multiple donors will be needed before this can be established (lines 540-546). However, we do mention in lines 558-562 of the 'Discussion' that the size of the GI tract might still be of importance.

-Line 502: There is some discussion in the literature of this kind:

'It has been shown that the infant gut actively recruits Bifidobacteria and Bacteroides thetaiotaomicrons by secretion of fucosylated oligosaccharides (FOS) into their gut lumen. (In: 'Symbiosis, dysbiosis, and rebiosis—The value

of metaproteomics in human microbiome monitoring. Lei Mao and Jacqueline Franke). In my opinion, this type of discussion should also be included here when talking about engraftment.

We thank the reviewer for bringing this publication to our attention. We have included a brief section in the 'Discussion' regarding host factors which might play a role in selecting for particular gut microbes (lines 562-570). However, since most of these findings relate to early life colonization in infants and due to a lack of clear information as to how host factors play a role in structuring the microbiota later on in life, we have refrained from elaborating too much on this subject.

-Line 520-523: This could be so but the fact that you only have one donor per age category makes this kind of argumentations difficult. If you had had 4 different donors of the same age but used to eat different diets, you could perhaps (probably) have found similar differences among the different donors. So, maybe it is more the individual variation (due to the many factors characteristics of each individual) than the age as a crucial factor.

This 'weakness' of the design should be more clearly emphasized.

We have removed these lines from the revised manuscript.

-Line 559-564: This refers to my comment above and I suggest that you move this text to the beginning of the discussion so that you tell the readers already there that you are aware of the limitations of the setup.

In our revised manuscript, we have greatly refrained from emphasizing the influence of donor age on colonization in our discussion. Additionally, we report the lack of donor replication in each age group as a drawback and a limitation of our study in the final paragraph of the discussion (lines 665-671). Since we no longer emphasize age as an important factor, and if the reviewer is also in agreement, we feel that this section doesn't need to be moved to the beginning of the 'Discussion'.

-Line 572-575: How was the performance of the pigs (and the mice)? Feed intake, body weight? Please add a couple of sentences in the beginning of the results reporting these data to indicate whether the animals thrived or deviated too much from 'normal' physiological conditions.

We did not measure any performance related parameters for either the piglets or the mice during this study. Although we were interested in collecting the initial and final body weights for the piglets, the complications of weighing piglets within the confines of a gnotobiotic isolator dissuaded us from doing so (especially when trying to get the initial body weights of newborn piglets). Past experiences in trying to weigh GF newborn piglets had shown us that this causes stress in these animals (who are very susceptible due to lack of immune development as a result of not getting chlostrum), and that if stressed too much at this early age can lead to their death. Furthermore, finding a suitable device to weigh pigs which can be sterilized properly and introduced into the isolator was a challenge, especially since we had to keep the piglets germ-free until weaning (4 weeks) prior to inoculation. Since animal performance wasn't of interest, we didn't want to risk any manipulation which could potentially compromise the sterile environment of the isolator. Since we didn't monitor performance for the piglets, we didn't do so for the mice either. The well-being of both species of animals was monitored through periodic visual observation by the attending veterinarians of the IACUC committee and daily

observations by other members of the IACUC staff. In addition, we dissected every piglet at euthanization to check for any aberrations in organ development which were not observed.

However, we do understand why the reviewer or anyone reading the manuscript would expect some information on animal performance if we were to make any statement regarding the influence of a rodent diet on growth of the piglets. In light of the fact that we are unable to provide such information, we have removed these sentences from the manuscript.

-Line 584: I suggest to add here or in the abstract your point on line 537 and onwards regarding the level at which the data are looked at. That is, maybe similarities are found at for ex. family level but it looks very different at a lower level.

This information has been added (lines 687-691).

-Line 591-594: how were the mice housed?

how many in each isolator?

how many isolators per donor?

how many isolators in total?

We have included this information in the updated 'methods' section (701-707).

-Line 601-602: How many isolators and piglets in total per donor and in total?

Is it 1 isolator per donor? That is, 4 isolators in total per animal species?

Yes, we had 1 isolator per donor per species so a total of 4 isolators per species. This information has been added to the 'Methods' section (lines 701-707; 719-724).

-Line 620-621: This is not clear. You should describe this procedure in more detail, explaining that you did a follow up study to look at variation among donorsWhat is the difference between the repeat study and the follow up study? This is unclear. I think you only had one follow up study where you used all 4 donors didn't you?.

Please describe more clearly your procedure.

Indeed we only had one follow-up study with a second set of INF, CHLD, ADLT, and SNR donors. We inoculated GF mice with all 4 of these donors but only the INF donor was transplanted into GF pigs as well. We used the term 'follow-up study' to refer to the transplantations done only in the mice (i.e. donors CHLD, ADLT, and SNR) and 'repeat study' to refer to the INF transplantations which were done in both pigs and mice. The idea was to emphasize that the 'repeat study' was to look at the repeatability of the results we had obtained for the first INF donor (which had established better in the mice compared to the piglets, which was opposite to what he had observed for the other 3 donors in the first study). We have modified the manuscript to make our thought process clearer (Lines 448 - 483).

-Line 639-640: when were the animals weaned and how many days after weaning were they inoculated?

The mice were weaned at 21 days of age and inoculated at 33 days of age.

The piglets were weaned at 28 days of age and inoculated at 31 days of age.

This information has been added to the manuscript (lines 766-768).

-Line 649: After second transplantation? After the first transplantation. We have clarified this in the manuscript (line 777).

-Line 696-698: By 'this analysis' you mean alpha diversity only? Please make it clear since you mention both alpha and beta diversity in the previous sentence.

Yes, only for alpha diversity. This has been clarified in this section (lines 833 – 835).

Reviewer #2 (Remarks to the Author):

Aluthge et al., described the impact of human microbiome on pig or mouse models to understand the microbiome implantation. As in most of the study are performed in mice models, thus authors took the advantage of the piglet model to explore whether it can be use for human nutritional and biomedical studies as many of anatomical, physiological and immunological similarity between human and mice. This unique study explored the direct engraftment of fecal bacterial from human donors of varying age range (infant, child, adult and elderly). Using 16S rRNA gene analysis using amplified sequence variants identified that more engraftment in piglet compared to the mice for child, adult and senior donors. But mice model is better off for the infant donor taxa. These results obtained in this study are novel and certainly interest to others in the community and the wider field.

I have some major and minor comments which could be helpful to improve the manuscript.

Major comments

1. Main concern of this study that authors used 16S rRNA sequence which does not allow to identify the functions of microbiome. Perhaps, authors should show the functional study using shotgun sequencing.

Although we agree that functional analysis is important, however, the main focus of the manuscript is engraftment of the donor bacterial taxa and persistence in the two animal models (which, as acknowledged by the reviewers, is an important gap in the current literature) and not the functional capacity. As such, we believe adding functional data is outside the scope of this manuscript. However, in the manuscript we used PICRUST to predict function of the microbiomes established (Supplementary Fig. 7) and demonstrate similar to the 16S rRNA based analysis the function favored the established taxa. As such our belief is if similar taxa is established similar function would be present (lines 510-513).

2. Both mice and piglet have different immune system may be due to these authors found the engraftment variability for child, adults and seniors rather than the size of the gut in the piglet which could accommodate more bacterial load.

We have referred to the differences between the porcine and murine immune system and the potential influence of the immune system in structuring the microbiota (lines 555 to 558). However, it is unclear whether immune system differences are the major factor contributing to the differences in colonization efficiency that we observe in the two animal models. It is likely that these differences are a result of multiple host factors (including the immune system), host genetics and environmental factors (such as diet) influencing the establishment of the microbiota.

3. The genera Bacteroides and Bifidobacterium colonize both animal models- is there functional similarity with the host genetics that's why both taxa are selected.

We're not aware of such a functional similarity at the level of host genetics which would explain why these taxa successfully established in both animal models.

4. Authors should another GF WT model (C57BL/6) to identify the genetics of mice does not affect the microbiome implantation.

This has been already reported and can be seen in "Comparative Evaluation of Establishing a Human Gut Microbial Community Within Rodent Models" by Melissa Wos-Oxley et al. (2012). (<https://pubmed.ncbi.nlm.nih.gov/22572831/>). Among the rodent models used in this study are mice of the C57BL/6 genetic background. Interestingly, similar to our observation with the C3H/HeN mice, these authors also observed inefficient colonization of Firmicutes members in the C57BL/6 mice as well.

Minor comments

1. Fig 1 and 3, poor quality of figures.

We have improved the quality of fig 1 & 3.

2. Do piglets have any cage effects?

No cage effects were observed in the study.

Additional changes on the revised manuscript

In addition to the above comments, we have performed an in-depth phylogenetic analysis to identify which Firmicutes colonized better in the pig model compared to the mouse model and show certain taxonomic groups favor colonization of the pig and some do not colonize both HMA species.

We defined a core set of ASVs that are present in the 3 mature microbiomes that went into both animal models and evaluated the establishment and colonization of these ASVs in the two HMA piglet models.

We have re-analyzed all data using the most recent version of the Silva database and have updated all taxonomic classifications.

Reviewers' Comments:

Reviewer #1:

Remarks to the Author:

Thank you for including the comments in the manuscript.

I only have a few comments listed below and also in the text itself:

-Line 107-109: You say that only one of the inoculum aliquots had sufficient material, so only 3 used for subsequent analysis. This is unclear to me, if only 1, how can you have 3?

-Line 736: Could you add the ingredient composition of the diet?

- Several places, added in the text as a comment, you write Firmicute instead of Firmicutes. Please correct.

- The quality and font size of the figures, especially Fig 5 and 6, should be improved. They are almost unreadable as they are now.

Reviewer #2:

Remarks to the Author:

Authors addressed my comments.

Reviewer #1 (Remarks to the Author):

Thank you for including the comments in the manuscript.

I only have a few comments listed below and also in the text itself:

Line 107-109: You say that only one of the inoculum aliquots had sufficient material, so only 3 used for subsequent analysis. This is unclear to me, if only 1, how can you have 3?

We apologize for having mis-typed what we intended to describe. Our intent was to state that for Donor_1, one of the 4 inoculum aliquots had insufficient material remaining for DNA extraction which resulted in only 3 of the aliquots being used for sequencing and subsequent analysis. This has been corrected in the manuscript as follows:

“For Donor_1, one of the inoculum aliquots used to inoculate the corresponding GF piglets did not have sufficient material remaining for DNA extraction, which resulted in only three inoculum aliquots from this donor for subsequent analysis.”

Line 736: Could you add the ingredient composition of the diet?

Upon reaching out to the company who produces this feed, we were informed that due to the proprietary nature of this commercial feed, they're unable to provide us the exact ingredient composition of the diet. Therefore, we're only able to provide the diet composition information provided in 'Supplementary Data 11'.

Several places, added in the text as a comment, you write Firmicute instead of Firmicutes. Please correct.

We thank the reviewer for pointing this out. The necessary corrections have been made.

The quality and font size of the figures, especially Fig 5 and 6, should be improved. They are almost unreadable as they are now.

We have improved the resolution and increased the font size of the figures.